# Dysfunctional hippocampal-prefrontal network underlies a multidimensional neuropsychiatric phenotype following early-life seizure

**Rafael Naime Ruggiero[1]\*, Danilo Benette Marques[1†],
Matheus Teixeira Rossignoli[1†], Jana Batista De Ross[1], Tamiris Prizon[1],
Ikaro Jesus Silva Beraldo[2,3], Lezio Soares Bueno-Junior[4], Ludmyla Kandratavicius[5],
Jose Eduardo Peixoto-Santos[6], Cleiton Lopes-Aguiar[2,3], Joao Pereira Leite[1]\***

[1]Department of Neuroscience and Behavioral Sciences, Ribeirão Preto Medical School, University of São Paulo, Ribeirão Preto, Brazil; [2]Department of Physiology and Biophysics Federal University of Minas Gerais, Belo Horizonte, Brazil; [3]Laboratory of Molecular and Behavioral Neuroscience (LANEC), Federal University of Minas Gerais, Belo Horizonte, Brazil; [4]Department of Psychiatry, University of Michigan Medical School, Ann Arbor, United States; [5]Department of Pathology, State University of Campinas, São Paulo, Brazil; [6]Neuroscience Discipline, Department of Neurology and Neurosurgery,Universidade Federal de São Paulo, São Paulo, Brazil

**\*For correspondence:**
rafaruggiero@gmail.com
(RNaimeR);
jpleite@fmrp.usp.br (JPereiraL)

[†]These authors contributed equally to this work

**Competing interest:** The authors declare that no competing interests exist.

**Abstract** Brain disturbances during development can have a lasting impact on neural function and behavior. Seizures during this critical period are linked to significant long-term consequences such as neurodevelopmental disorders, cognitive impairments, and psychiatric symptoms, resulting in a complex spectrum of multimorbidity. The hippocampus-prefrontal cortex (HPC-PFC) circuit emerges as a potential common link between such disorders. However, the mechanisms underlying these outcomes and how they relate to specific behavioral alterations are unclear. We hypothesized that specific dysfunctions of hippocampal-cortical communication due to early-life seizure would be associated with distinct behavioral alterations observed in adulthood. Here, we performed a multi-level study to investigate behavioral, electrophysiological, histopathological, and neurochemical long-term consequences of early-life *Status epilepticus* in male rats. We show that adult animals submitted to early-life seizure (ELS) present working memory impairments and sensorimotor disturbances, such as hyperlocomotion, poor sensorimotor gating, and sensitivity to psychostimulants despite not exhibiting neuronal loss. Surprisingly, cognitive deficits were linked to an aberrant increase in the HPC-PFC long-term potentiation (LTP) in a U-shaped manner, while sensorimotor alterations were associated with heightened neuroinflammation, as verified by glial fibrillary acidic protein (GFAP) expression, and altered dopamine neurotransmission. Furthermore, ELS rats displayed impaired HPC-PFC theta-gamma coordination and an abnormal brain state during active behavior resembling rapid eye movement (REM) sleep oscillatory dynamics. Our results point to impaired HPC-PFC functional connectivity as a possible pathophysiological mechanism by which ELS can cause cognitive deficits and psychiatric-like manifestations even without neuronal loss, bearing translational implications for understanding the spectrum of multidimensional developmental disorders linked to early-life seizures.

## eLife assessment

This **important** study assesses anatomical, behavioral, physiological, and neurochemical effects of early-life seizures in rats, describing a striking astrogliosis and deficits in cognition and electrophysiological parameters. The **solid** results come from a wide range of convergent techniques that were used to understand the effects of early-life seizures on behavior as well as hippocampal prefrontal cortical dynamics. This paper will be of interest to neurobiologists, epileptologists, and behavioral scientists.

## Introduction

The early years of life represent a critical period for neuronal maturation and the development of cognitive functions (*Short and Baram, 2019*). Disturbances during this period can have a lasting impact on neuronal activity and cause multifaceted neuropsychiatric manifestations (*Marín, 2016*). Seizures are a prevalent disturbance in the immature brain and are often associated with profound and enduring adverse neurological outcomes that persist throughout adulthood (*Brunquell et al., 2002*; *Holmes, 2009*). Children who experience early-life seizures (ELS) not only face an increased likelihood of developing epilepsy but also have a heightened susceptibility to cognitive impairments and behavioral disorders, including autism spectrum disorder (ASD), attention deficit hyperactivity disorder (ADHD), intellectual disability, and psychiatric symptoms like schizophrenia (*Dreier et al., 2019*; *Holmes, 2009*; *Kaufmann et al., 2009*). Since the pathophysiological mechanisms linking ELS and neurodevelopmental disorders are complex, understanding how ELS per se affects neuronal function and dissecting the associations between neurobiological and behavioral alterations are crucial for developing better treatments for these conditions.

In experimental models, ELS induces long-term cognitive and behavioral alterations with translational validity with the main neuropsychiatric domains affected in humans (*Holmes, 2016*). Studies have reported that a prolonged ELS in rodents can lead to cognitive deficits, increased anxiety, reduced social interaction, and facilitated development of epilepsy (*Castelhano et al., 2013*; *de Oliveira et al., 2008*; *Kubová et al., 2004*). Interestingly, ELS models do not exhibit severe neuronal loss or structural damage (*Baram et al., 2002*; *Raol et al., 2003*), suggesting that the behavioral alterations induced by ELS may be mediated by functional and molecular effects at the brain circuit level (*Kandratavicius et al., 2012b*; *Kandratavicius et al., 2012a*). The hippocampal-prefrontal cortical circuit (HPC-PFC) plays a crucial role in cognitive and sensorimotor processes, and evidence converges to indicate its dysfunction as a possible common mechanism shared between ELS, neurodevelopmental, and psychiatric disorders, especially epilepsy, psychosis/schizophrenia, ASD, and cognitive disability (*Ruggiero et al., 2021*; *Sigurdsson and Duvarci, 2016b*). Reports indicate that ELS impacts HPC-PFC functional connectivity, as revealed by the coordination of network oscillations during particular situations (*Barry et al., 2020*; *Kleen et al., 2011*; *Mouchati et al., 2019*; *Niedecker et al., 2021*). However, how specific functional or molecular effects of ELS on the HPC-PFC network are associated with separate behavioral dimensions within this complex multimorbidity spectrum remains unknown. Also, since behavioral alterations of ELS are long-lasting and progressive, a comprehensive characterization of network dynamics across brain states would be pivotal for a complete understanding of the effects of ELS on brain function.

Here, we hypothesized that ELS would produce long-term alterations in HPC-PFC neuronal dynamics related to behavioral abnormalities (*Ruggiero et al., 2012*). To test this hypothesis, we induced *Status epilepticus* (SE) as a model of prolonged ELS and performed behavioral tests to assess cognitive and sensorimotor processes. Then, we performed neurophysiological, immunohistochemical, and neurochemical experiments to study potential changes in the HPC-PFC circuits that could underlie such behavioral alterations. We found that adult ELS rats present a myriad of behavioral impairments: working memory deficits, hyperlocomotion, poor sensorimotor gating, and behavioral sensitivity to psychostimulants. Surprisingly, we found an enhanced long-term potentiation (LTP) in the HPC-PFC pathway of ELS animals correlated with cognitive impairment in a U-shaped manner. Our results also describe neuroinflammation and alterations in dopaminergic neurotransmission as significant contributors to sensorimotor disturbances. We also investigated HPC-PFC dynamics across different brain states in freely-moving rats. ELS rats displayed an increased HPC theta power and impaired coordination of PFC gamma activity during active behavior. During rapid eye movement

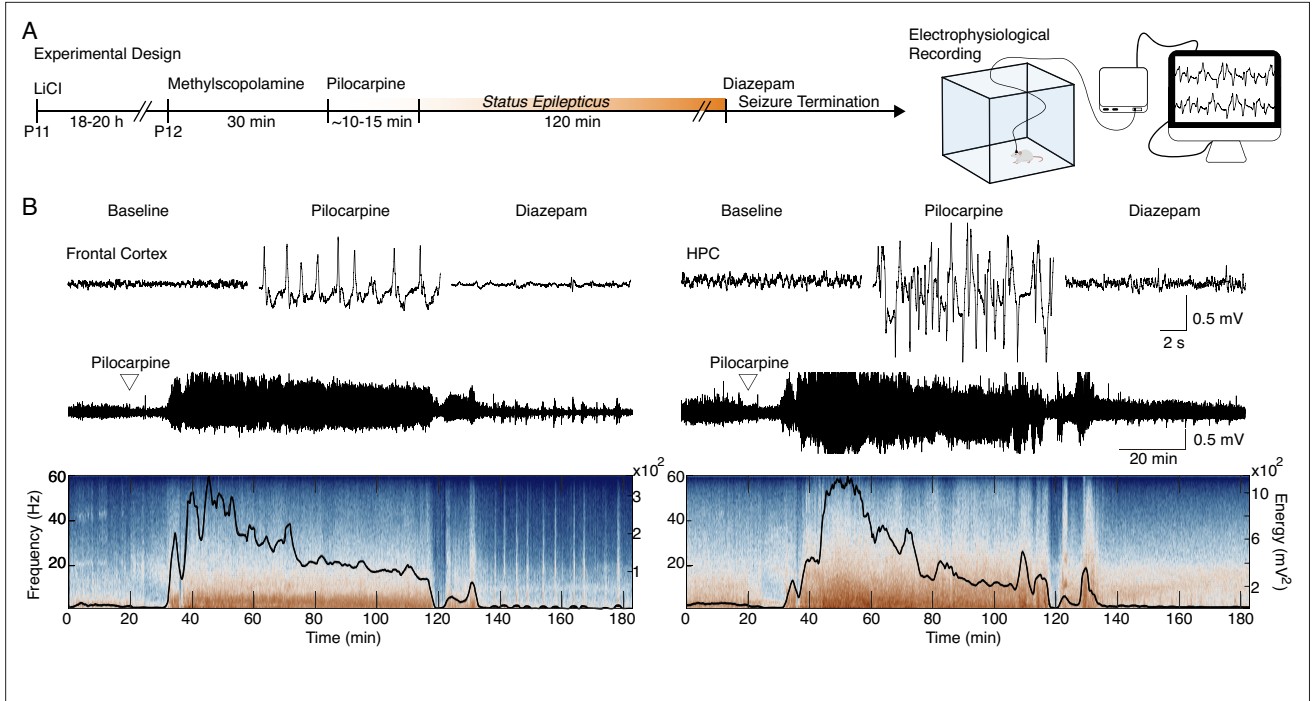

**Figure 1.** Early-life seizure (ELS) induction. (**A**) Experimental design representing early-life *Status epilepticus* (SE) induction and monitoring. (**B**) Representative traces of local field potential (LFP) during baseline, motor seizure induced by pilocarpine, and ELS interruption with diazepam injection (top). Representative LFP from the frontal cortex and HPC throughout the experiment (middle). LFPs are aligned with their respective spectrograms and energy curves (bottom). ELS n=5.

(REM) sleep, ELS rats showed a reduction in HPC-PFC coherence. Remarkably, ELS rats displayed an aberrant brain state during active behavior characterized by oscillatory dynamics oddly similar to REM sleep. Finally, we implemented a linear discriminant model, demonstrating that collective patterns of oscillatory alterations accurately distinguished ELS from control animals. Interestingly, although these alterations relate to broad clinical manifestations of neurodevelopmental disturbances and psychiatric manifestations, we recognize a particular convergence of dysfunctional patterns at multiple levels linking the consequences of ELS to functional alterations reported in animal models of epilepsy, psychosis/schizophrenia, ASD, and cognitive disability. These convergent patterns indicate that epileptogenic processes triggered during early life, such as neuroinflammation, could lead to altered neurotransmission and, consequently, aberrant dynamics in the HPC-PFC coordination, potentially contributing to the neurological and psychiatric manifestations linked to early-life seizure.

## Results

### ELS induces cognitive and behavioral impairments in adulthood

SE induced by lithium-pilocarpine in early life (*Figure 1A*) produced an unequivocal and stereotyped seizure, consisting mainly of forelimb myoclonus and orofacial automatism (Racine 3), with some animals showing further development into bilateral forelimb myoclonus, rearing and falling (Racine 4 and 5; *Racine, 1971*). Seizures were confirmed electrophysiologically in freely moving recordings of the frontal cortex and HPC, with paroxysms starting 10 min after pilocarpine injection and persisting for at least 120 min (*Figure 1B*).

Next, we hypothesized that ELS rats would display a wide range of long-term cognitive and sensorimotor consequences. Therefore, we submitted the animals to several behavioral tests. (*Figure 2A*). First, we evaluated possible weight differences as a confounder of behavioral measures. ELS rats displayed weight loss on the day after SE (Two-way RM ANOVA, treatment effect: $F_{(9,207)}$ = 11.7, p<0.0001, Sidak's *post-hoc:* p<0.0001; *Figure 2B*). However, ELS animals recovered the pace of normal weight gain in the subsequent days (*post-hoc:* p>0.05) and into adulthood (*Figure 2C*),

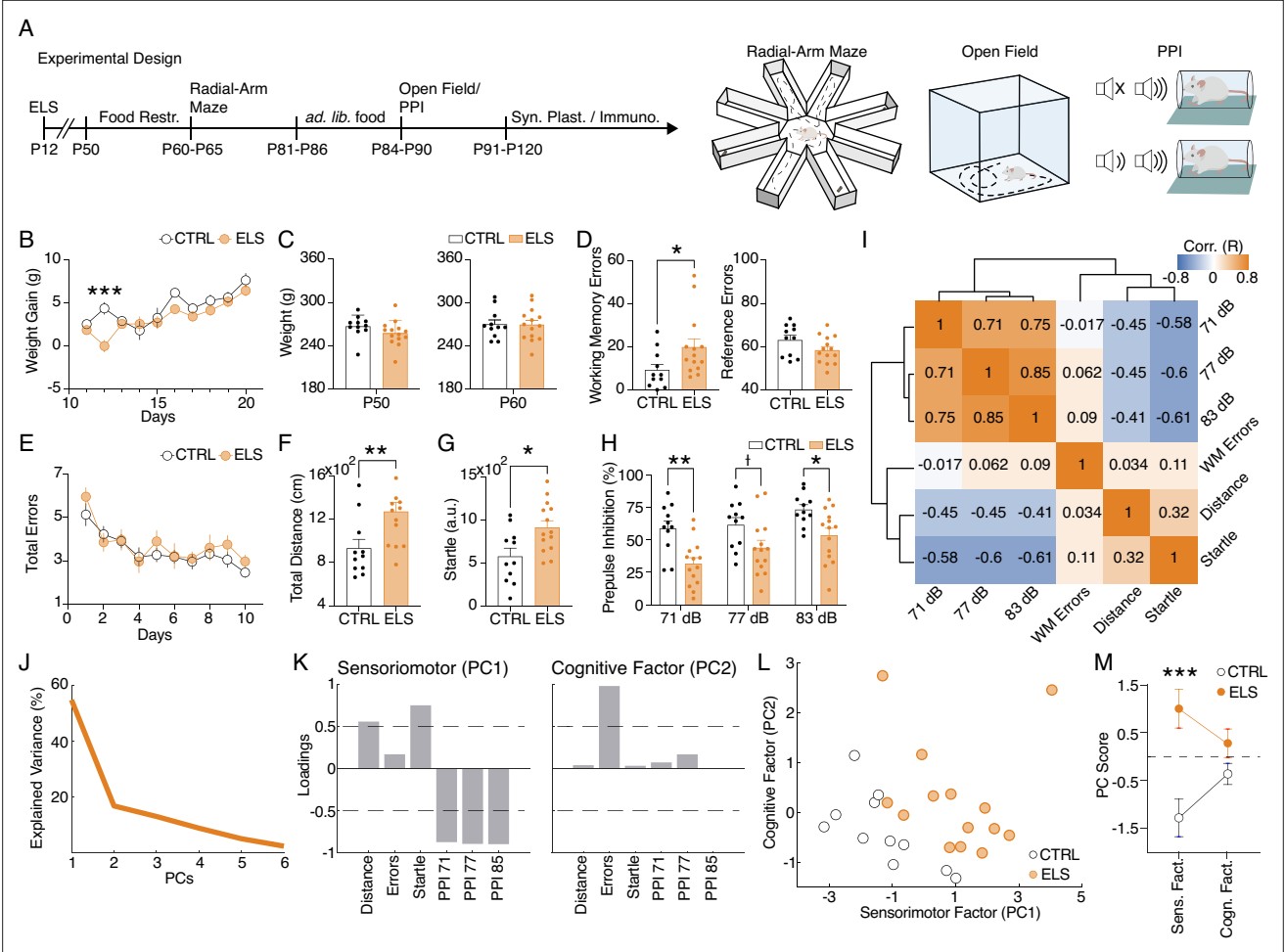

**Figure 2.** Early-life seizure (ELS) induces cognitive and sensorimotor impairments. (**A**) Experimental design representing early-life *Status epilepticus* (SE) induction, age of the animals during the experiments, and behavioral tests performed. (**B**) Weight gain after ELS induction. ELS rats showed weight loss only on the ELS induction day (Two-way RM ANOVA). (**C**) No weight differences between ELS and control (CTRL) throughout experiments in adulthood, regardless if before food restriction (left) or during food restriction (right). (**D**) Working memory task on the radial-arm maze. ELS rats showed an increase in working memory errors across sessions (left, Mann-Whitney test) without differences in the number of reference errors (right). (**E**) ELS rats showed a similar learning curve when compared to CTRL. (**F**) Hyperlocomotion of ELS rats in the open field test (Mann-Whitney test). (**G**) Higher acoustic startle of ELS rats (*t*-test). (**H**) Impaired sensorimotor gating in ELS rats as measured in the prepulse inhibition (PPI) test with three stimulus intensities (Two-way RM ANOVA). (**I**) Correlation matrix of all behavioral variables. Note a cluster of positive correlations formed by the PPI measures and another cluster of negative correlations formed by the PPI measures with distance traveled on the open field and the acoustic startle. Working memory errors were not correlated with the other behavioral features. (**J**) Explained variance from principal components analysis (PCA). Note that the two first PCs explain 71.32% of the data. (**K**) Loadings of the behavioral variables on each principal component (PC) reveal a *sensorimotor factor* (PC1) and an uncorrelated *cognitive factor* (PC2). (**L**) PCA of the behavioral features showing each animal projected onto the new factors. (**M**) The sensorimotor factor distinguishes ELS from CTRL animals, as we compare the PC scores from each group (*t*-test). † p<0.1, *p<0.05, **p<0.01, ***p<0.001. CTRL n=11, and ELS n=14. Error bars represent the mean ± SEM.

showing no distinction to control group (CTRL) before (*t*-test, $t_{(23)}$ = 1.440, p=0.1634) or during food restriction (*t*-test, $t_{(23)}$ = 0.0535, p=0.9577), indicating that behavioral effects were unrelated to weight differences or alterations in food-seeking.

Importantly, when examined for spatial memory in the radial-arm maze, ELS rats presented a specific increase in working memory errors (Mann-Whitney test, *U*=36.5, *n(CTRL, ELS)*=11, 14, p=0.0254; *Figure 2D*). We found no differences in the total reference errors (*t*-test, $t_{(23)}$ = 1.719, p=0.0991; *Figure 2D*) and in the learning curve (Two-way RM ANOVA, treatment effect: $F_{(1,23)}$ = 0.8032, p=0.3794; *Figure 2E*). In addition, ELS rats exhibited significantly higher locomotion in the open field test (Mann-Whitney test, *U*=27, *n(CTRL, ELS)*=11/14, p=0.0051; *Figure 2F*). ELS rats also displayed a higher startle response (*t*-test, $t_{(23)}$ = 2.757, p=0.011; *Figure 2G*) and a deficit in sensorimotor gating

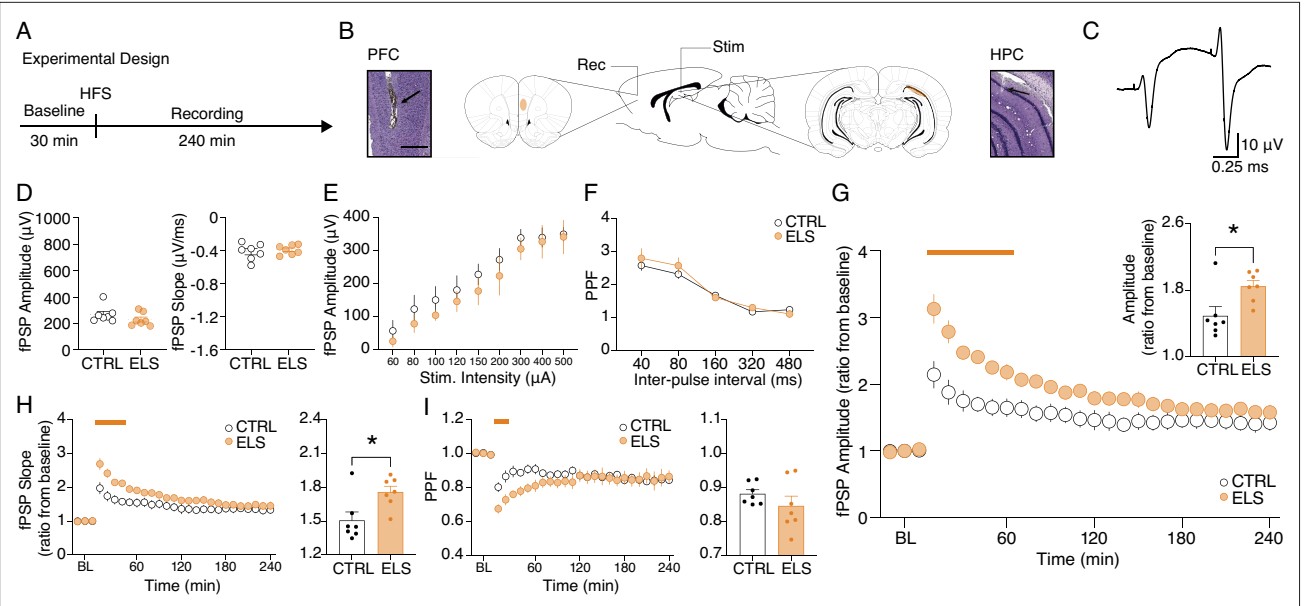

**Figure 3.** Aberrant hippocampus-prefrontal cortex (HPC-PFC) synaptic plasticity in animals submitted to early-life seizure (ELS). (**A**) Experimental design including high-frequency stimulation (HFS) to long-term potentiation (LTP) induction. (**B**) Electrode placement and representative electrolytic lesions in Cresyl Violet stained coronal sections. (**C**) Representative evoked field postsynaptic potentials (fPSPs) in the PFC. No difference between ELS and control (CTRL) rats in terms of basal neurotransmission and synaptic efficacy, as shown by: (**D**) fPSP amplitude (left) and slope (right); (**E**) input-output curve; and (**F**) paired-pulse facilitation (PPF) curve.( **G**) Despite no changes in basal synaptic efficacy, ELS rats showed aberrantly increased LTP, represented by higher fPSP amplitude ratios from baseline (BL) post-HFS, both across time blocks (main curve graph, Two-way RM ANOVA) and their averages (bar graph, inset, *t*-test). (**H**) As in (**G**) but from fPSP slopes. (**I**) ELS rats showed reduced PPF in the initial 30 min (curve graph). Averaged data resulted in no differences between the groups (bar graph, inset). † p<0.1, *p<0.05, **p<0.01, ***p<0.001. CTRL n=7, and ELS n=7. Error bars represent the mean ± SEM.

represented by a reduction in prepulse inhibition (PPI) of the acoustic startle (Two-way RM ANOVA, treatment effect: $F_{(2,46)}$ = 19.6, p<0.0001, Sidak's *post-hoc:* PPI 71 dB: p=0.0017; PPI 77 dB: p=0.1040; PPI 83 dB: p=0.030; *Figure 2H*).

Interestingly, all these behavioral alterations are described in animal models of neurodevelopmental disorders such as ASD and schizophrenia (*Ey et al., 2011*; *Scott et al., 2018*; *Sigurdsson, 2016a*). Then, we investigated if collective patterns of behavioral alterations induced by ELS could reflect latent factors representing more comprehensive behavioral dimensions. Working memory errors in the radial-arm maze showed little correlation with the other behavioral variables, while sensorimotor variables (locomotion, startle response, and sensorimotor gating) presented a strong correlation with each other. PPI exhibited a positive correlation with every intensity and a negative correlation with startle response and locomotion (*Figure 2I*). We also performed principal component analysis (PCA), to extract patterns of variance between multivariate behavioral data (*Figure 2J–M*). Our analysis indicated that dimensionality reduction by PCA using the first two components explained 71.32% of the variability of the data (*Figure 2J*). By analyzing the loadings of the PC1, the variables that contribute more strongly (>0.5) to this dimension were: high total distance, startle reflex, and low PPI scores, indicating this component represents a latent factor of sensorimotor dysfunction. In turn, the PC2 represents the cognitive dimension of the behavioral variables, as it presents a high loading value (>0.5) for working memory errors (*Figure 2K*). Finally, we found that these PCs clearly distinguished CTRL from ELS animals, which presented higher values of PC1 (PC1: *t*-test, $t_{(23)}$ = 3.9713, p=0.0006; PC2: *t*-test, $t_{(23)}$ = 1.6466, p=0.1132; *Figure 2L–M*). These results indicate that the covariance across behavioral tests can be summarized by a latent *sensorimotor factor* (PC1) and a *cognitive factor* (PC2), representing two distinct dimensions of the ELS-induced behavioral phenotype and suggesting possibly different neurobiological bases.

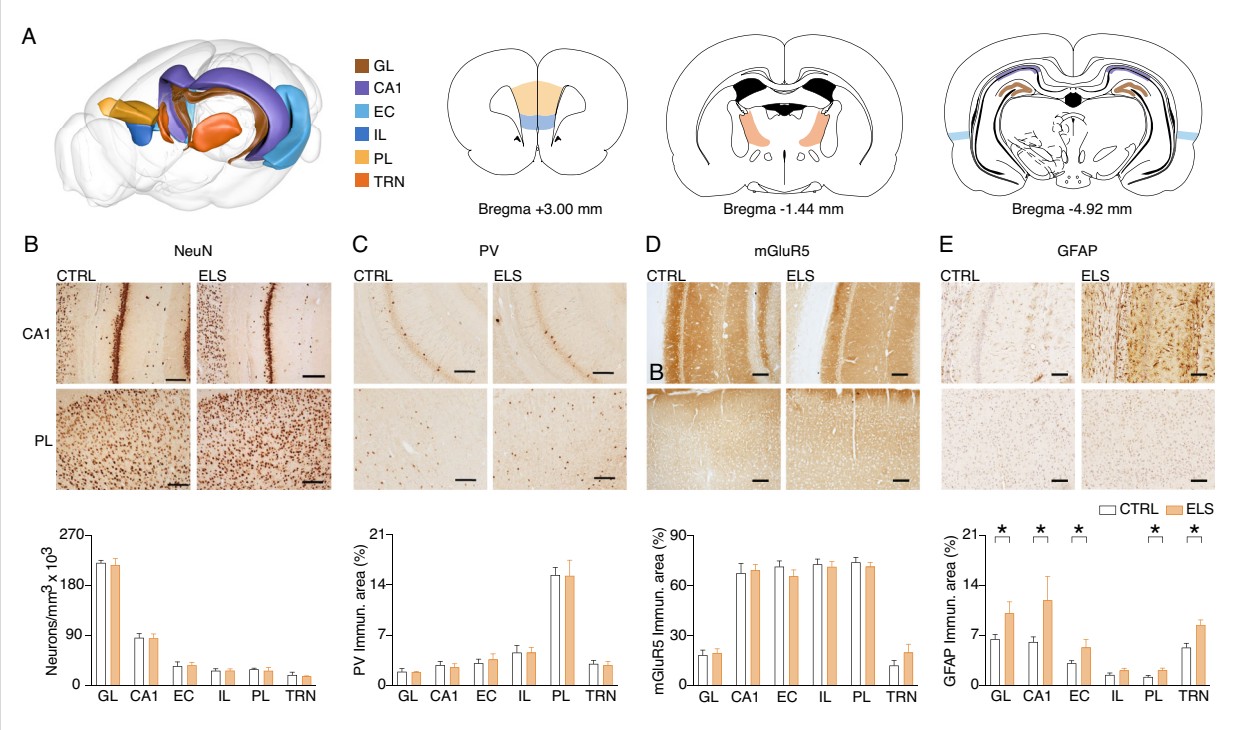

**Figure 4.** Early-life seizure (ELS) induces long-term neural inflammation, but not neuronal loss. (**A**) 3-D brain diagram and coronal sections representing the brain regions investigated. (**B–E**) Representative photomicrographs from immunohistochemistry each pair of columns represents a group comparison (left control (CTRL) and right ELS) and each line is related to CA1 and PL, respectively. (**B**) NeuN immunohistochemistry showed that ELS rats did not present neuronal loss. Parvalbumin (PV) immunohistochemistry. (**D**) mGluR5: (**E**) GFAP, showing increased astrogliosis induced by ELS (Mixed-effects model). GL: Granule layer of dentate gyrus; CA1: *Cornu Ammonis* 1; EC: Entorhinal cortex; IL: Infralimbic cortex; PL: Prelimbic cortex; TRN: Thalamic Reticular Nucleus. *p<0.05. CTRL n=11, and ELS n=14. Error bars represent the mean ± SEM.

## Aberrant HPC-PFC synaptic plasticity in animals submitted to ELS

Next, we tested the hypothesis that behavioral abnormalities in ELS rats could be associated with synaptic plasticity impairments of the HPC-PFC pathway (*Figure 3A–C*). First, we looked for ELS-related alterations in basal synaptic efficacy before inducing LTP. Synaptic efficacy was assessed in terms of basal field postsynaptic potential (fPSP) amplitude and slope (amplitude: *t*-test, $t_{(12)} = 1.278$, p=0.2255; slope: *t*-test, $t_{(12)} = 0.5470$, p=0.5944; *Figure 3D*), input-output curve (fPSP amplitude as a function of stimulus intensity) (Two-way RM ANOVA, treatment effect: $F_{(1,12)} = 0.6596$, p=0.4325; *Figure 3E*) and paired-pulse facilitation (PPF) curve (PPF as a function of the inter-pulse interval) (Two-way RM ANOVA, treatment effect: $F_{(1,12)} = 0.3345$, p=0.5737; *Figure 3F*). None of these measures indicated alterations in ELS when compared to CTRL (*Figure 3D–F*).

Then, we proceeded to the synaptic plasticity experiments and found that ELS rats presented an abnormally augmented LTP compared to CTRL (*Figure 3G*; Two-way RM ANOVA, treatment effect: $F_{(1,12)} = 7.54$, p=0.0177; interaction effect: $F_{(26,312)} = 7.393$, p<0.0001; Sidak's *post-hoc*: p<0.05 for 10–60 min; *t*-test, $t_{(12)} = 2.747$, P=0.0177; *Figure 3G* inset), despite the lack of alterations in basal synaptic efficacy described above (*Figure 3D–F*). Aberrant LTP in the ELS rats was also observed when analyzing the fPSP slope (Two-way RM ANOVA, treatment effect: $F_{(1,12)} = 4.757$, p=0.0498; interaction effect: $F_{(26,312)} = 4.120$, p<0.0001; Sidak's *post-hoc*: p<0.05 for 10–40 min; *Figure 3H*; *t*-test, $t_{(23)} = 2.794$, p=0.0162; *Figure 3H* inset). However, short-term plasticity measured by PPF decreased during the initial 30 min of recording (Two-way RM ANOVA, interaction effect: $F_{(26,312)} = 3.457$, p<0.0001; Sidak's *post-hoc*: p<0.05 for 20–30 min; *Figure 3I*), possibly reflecting a ceiling effect on fPSP amplitude after LTP induction. With these results, we demonstrate that, although ELS does not affect basal synaptic efficiency, it significantly increases the ability of HPC-PFC circuits to undergo LTP.

## ELS induces long-term neural inflammation, but not neuronal loss

We sought to investigate histopathological alterations in cortico-limbic circuits that could underlie the abnormalities in behavior and HPC-PFC synaptic plasticity of ELS rats (*Figure 4A*). In accordance with previous findings, ELS rats did not present neuronal loss when adults, as verified by NeuN expression (Mixed-effects model, $F_{(1,12)}$ = 0.7683, p=0.3980; uncorrected Mann-Whitney test, p=0.1666–0.9372; *Figure 4B*). However, we investigated potential immunohistochemical alterations associated with epilepsy and neuropsychiatric disorders and their potential correlation with functional changes. Specifically, our investigation focused on assessing the status of parvalbumin (PV) interneurons, mGluR5 expression, and glial fibrillary acidic protein (GFAP) in the hippocampus CA1 and granule layer (GL) of the dentate gyrus, entorhinal cortex (EC), infralimbic (IL) and prelimbic cortex (PL), and thalamic reticular nucleus (TRN) as potential indicators of underlying pathophysiological mechanisms in hippocampo-thalamic-cortical circuits. ELS did not produce long-term effects in the expression of PV (Mixed-effects model, $F_{(1,13)}$ = 0.008, p=0.9297; uncorrected Mann-Whitney test, p=0.3659–0.9307; *Figure 4C*) or mGluR5 receptors (Mixed-effects model, $F_{(1,11)}$ = 0.0011, p=0.9735; uncorrected Mann-Whitney test, p=0.2467–0.8181; *Figure 4D*). Finally, we found a strong increase in neuroinflammation, verified by the marked expression of GFAP in ELS rats in all regions investigated except the infralimbic cortex (Mixed-effects model, $F_{(1,18)}$ = 5.368, p=0.0325; uncorrected Mann-Whitney test, p=0.0067–0.0419 for GL, CA1, EC, PL, and TRN, and p=0.2298 for IL; *Figure 4E*). First, our observations confirm that relevant cognitive and behavioral impairments can manifest after ELS even in the absence of substantial neuronal loss in the investigated brain areas. Meanwhile, these results show that ELS produces long-term and widespread brain inflammation. Noteworthy, the neuroinflammation promoted by SE extends beyond the HPC-PFC circuit, also affecting thalamic nuclei, suggesting a large-scale impact.

## Cognitive impairment and sensorimotor disturbances are distinctly correlated with abnormal LTP and neuroinflammation

So far, we have found that ELS promotes sensorimotor and cognitive behavioral alterations parallel to inducing exaggerated HPC-PFC LTP and large-scale neuroinflammation while exhibiting no significant neuronal loss and subtle molecular changes. However, we still do not know how specific neurobiological alterations relate to particular behavioral changes. By understanding the association between behavioral and neurobiological variables, we would be able to infer putative mechanisms underlying the behavioral comorbidities observed in ELS animals. In this context, because neurodevelopmental insults can have multifaceted and intricate outcomes, we also sought to explore multivariate and non-linear associations within our data.

To investigate multivariate relationships between the behavioral (cognitive and sensorimotor factors) and neurobiological (synaptic plasticity and immunohistochemical measures) dimensions of ELS effects, we conducted canonical correlation analysis (CCA) (*Figure 5A*). CCA produced a significant high-correlation model (r=0.94, Hotteling-Lawley trace test, *F-approximation*$_{(10,12)}$=3.2311, p=0.0164; *Figure 5B*) that clearly separated CTRL from ELS (behavior canonical dimension, *t*-test, $t_{(11)}$ = 3.693, p=0.0035; neurobiology canonical dimension, *t*-test, $t_{(11)}$ = 3.917, p=0.0024; *Figure 5C*). By analyzing the coefficients, we observed that the behavior canonical dimension showed higher values for the sensorimotor factor, but also included the cognitive factor, although at a lower degree (*Figure 5D*). The neurobiology canonical dimension showed a larger value for the GFAP, intermediate values for LTP and PV and the lowest coefficients for NeuN and mGluR5 (*Figure 5D*). A similar pattern was observed when analyzing the loadings for each dimension (*Figure 5E*). Notably, the neurobiological variables have only weak correlations between them (*Figure 5F*). To further dissect specific multivariate relationships between behavioral and neurobiological measures, we constructed generalized linear models (GLM). The GLM indicates GFAP, PV, and LTP as the most significant variables that explain the sensorimotor factor variance coefficients = 2.3775, 0.8457, and 0.5187; p=0.1164, 0.00548, 0.08474 for GFAP, PV, and LTP, respectively, (*Figure 5G*). Accordingly, GLM multivariate regression indicates a strong association between the sensorimotor factor and all significant GLM coefficients (GFAP, PV, plus LTP: $r^2$=0.7974, p=0.0116; *Figure 5H*), as well as sensorimotor factor and univariate neurobiological measures (GFAP: r=0.6840, p=0.0099; *Figure 5I*, PV: r=0.6548, p=0.0151; *Figure 5J*).

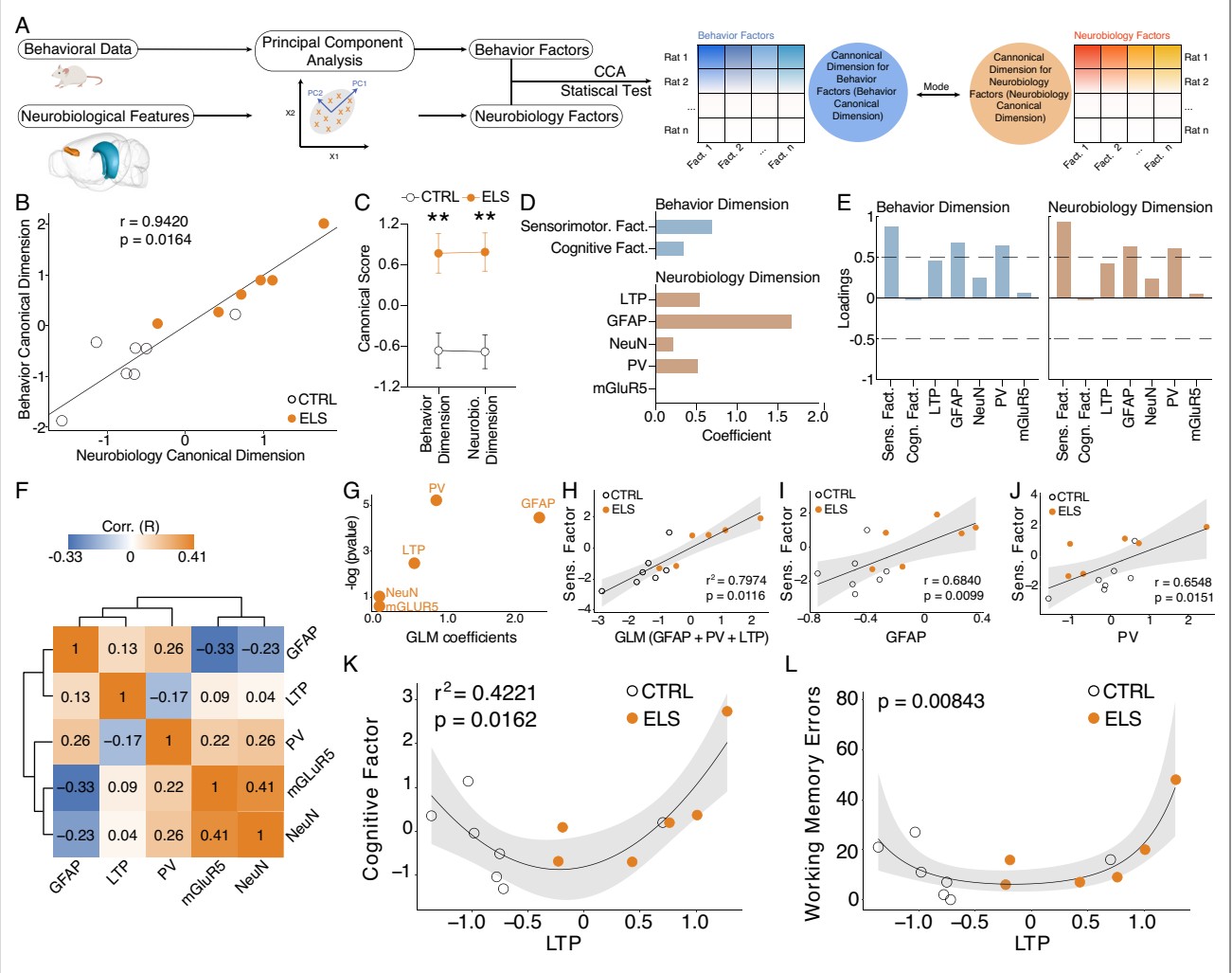

**Figure 5.** Cognitive impairment and sensorimotor deficits are distinctly correlated with abnormal long-term potentiation (LTP) and neuroinflammation. (**A**) Steps performed in data processing and analysis for canonical correlation analysis (CCA). (**B**) CCA analysis showing control (CTRL) and early-life seizure (ELS) animals projected onto the first canonical correlation. We found a strong correlation between the neurobiology canonical dimension and the behavior canonical dimension (Hotteling-Lawley trace test). (**C**) Canonical scores showing the clear CTRL *vs.* ELS difference that the CCA was able to capture (*t*-test). (**D**) Coefficients for behavior and neurobiology canonical dimensions, showing greater contributions of the sensorimotor factor, glial fibrillary acidic protein (GFAP), parvalbumin (PV), and LTP for the correlation. (**E**) Loadings of the behavioral and neurobiological variables on each canonical correlation dimension indicating an association between GFAP and PV with the sensorimotor factor. (**F**) Correlation matrix showing only weak correlations between the neurobiological features. (**G**) Generalized linear model (GLM) regression between neurobiological features and the sensorimotor factor. (**H**) GLM model between neurobiological features and the sensorimotor factor show a strong association with GFAP, PV, and LTP. (**I**) GLM regression between LTP and the cognitive factor shows a quadratic association (left). This quadratic association was further confirmed by performing a quasi-Poisson regression between LTP and raw values of working memory errors (right). \*\*p<0.01. CTRL n=7, and ELS n=6. Error bars represent the mean ± SEM.

Strikingly, we found a quadratic relationship between synaptic plasticity effects and measures of cognitive impairment. Poor performance on the working memory test was observed in rats that presented too low or too high values of HPC-PFC LTP, suggesting a U-shaped relationship between LTP and cognitive performance (*Figure 5K–L*). We observed this quadratic relationship using both the cognitive factor (gaussian GLM, LTP coefficient = 1.22, p=0.0162; *Figure 5K*) and raw error values from the radial maze test using a quasi-Poisson GLM (p=0.00843; *Figure 5L*).

Taken together, our analysis demonstrates that both neuroinflammation and synaptic plasticity alterations may underlie the behavioral effects of ELS, with neuroinflammation being more associated with the sensorimotor dimension and LTP with the cognitive dimension.

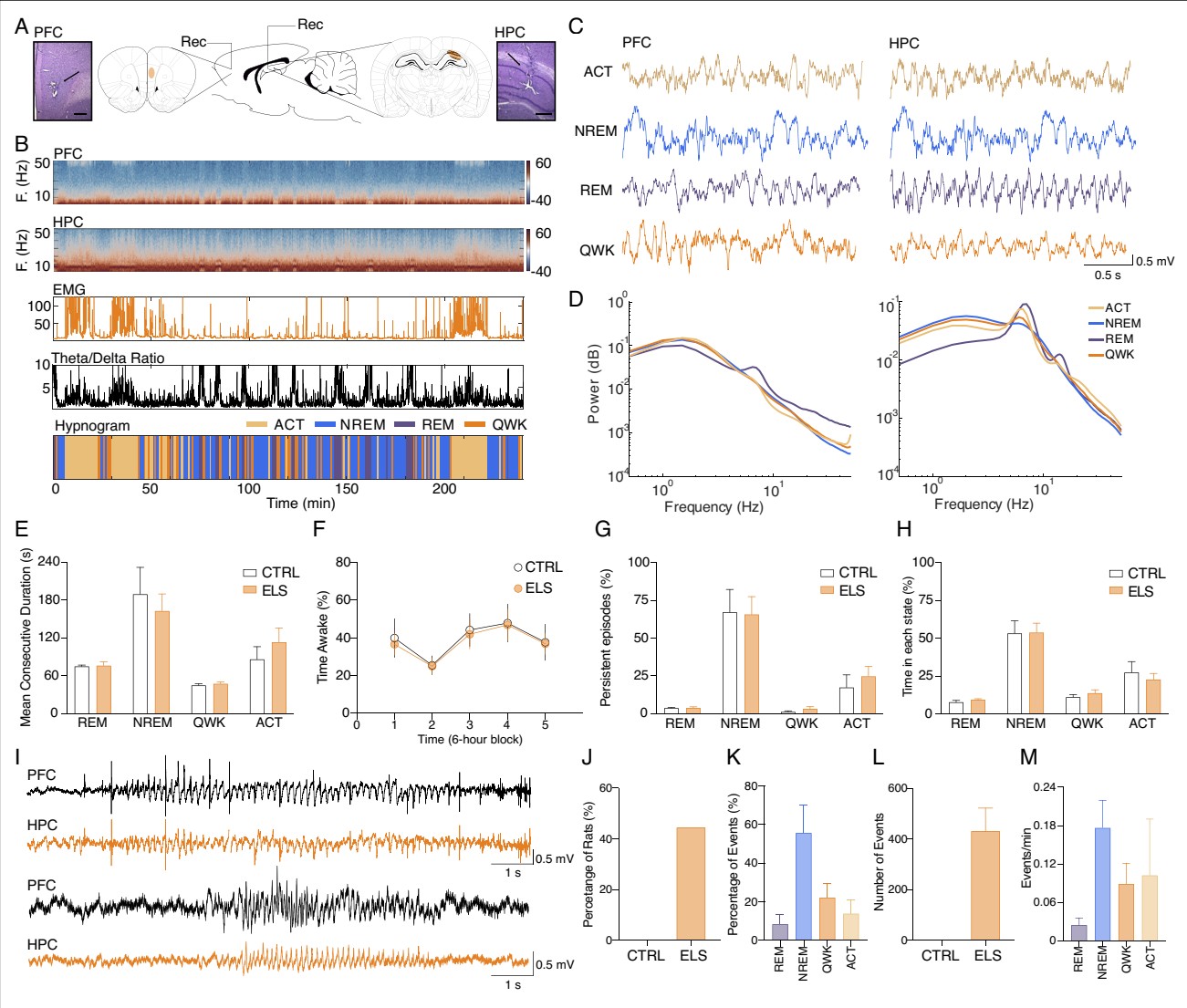

**Figure 6.** Classification of sleep-wake states and electrographic discharges in early-life seizure (ELS) rats. (**A**) Electrode placement and representative electrolytic lesions in Nissl-stained coronal sections. (**B**) Representative spectrogram, electromyogram (EMG) root mean squared (RMS), and HPC theta/delta ratio used to score the sleep-wake cycle and to generate the hypnogram (bottom). (**C**) Representative local field potentials (LFPs) (top) recorded from PFC and HPC during active wake (ACT), non-REM (NREM), rapid eye movement (REM), and quiet wake (QWK). (**D**) Average power spectral density for each state in the PFC and HPC of a representative control (CTRL) animal. Note the increase in HPC theta power during ACT and REM. ELS rats did not present alterations in the sleep-wake cycle. (**E–H**) There were no differences between ELS and CTRL in the (**E**) Mean duration of consecutive epochs for each state, (**F**) percentage of time awake across 6 hr blocks, and in the (**G**) number of persistent episodes (episodes lasting longer than 3 s, top) or the (**H**) percentage of time spent in each state (bottom). (**I**) Representative LFP from PFC and HPC showing a representative spike-wave discharge (SWD, top) and a poly-spike discharge (bottom). (**J**) Electrographic discharges were found in 44.5% of ELS animals and were absent in CTRL animals. (**K**) Percentage of events occurring in each sleep-wake stage. (**L**) Average total number of SWD events in CTRL and ELS animals. (**M**) Frequency of SWDs occurring per state. SWDs events rarely occurred during REM. CTRL n=6, and ELS n=9. Error bars represent the mean ± SEM.

## ELS animals present epileptiform-like activity when adults

Given the results above indicating ELS impacts on specific behavioral demands, large-scale networks, and HPC-PFC functional connectivity, we decided to investigate ELS-related dysfunctions in circuit dynamics across different brain-behavioral states using chronic electrophysiology recordings. We hypothesized that HPC-PFC communication is altered in ELS animals, showing aberrant oscillatory patterns in the circuit. First, we classified local field potentials (LFP) epochs into active wake (ACT), REM, non-REM (NREM) sleep, and quiet wake (QWK; *Figure 6A–D*). No CTRL *vs.* ELS differences were found in sleep architecture, according to the mean consecutive duration of each state, percentage

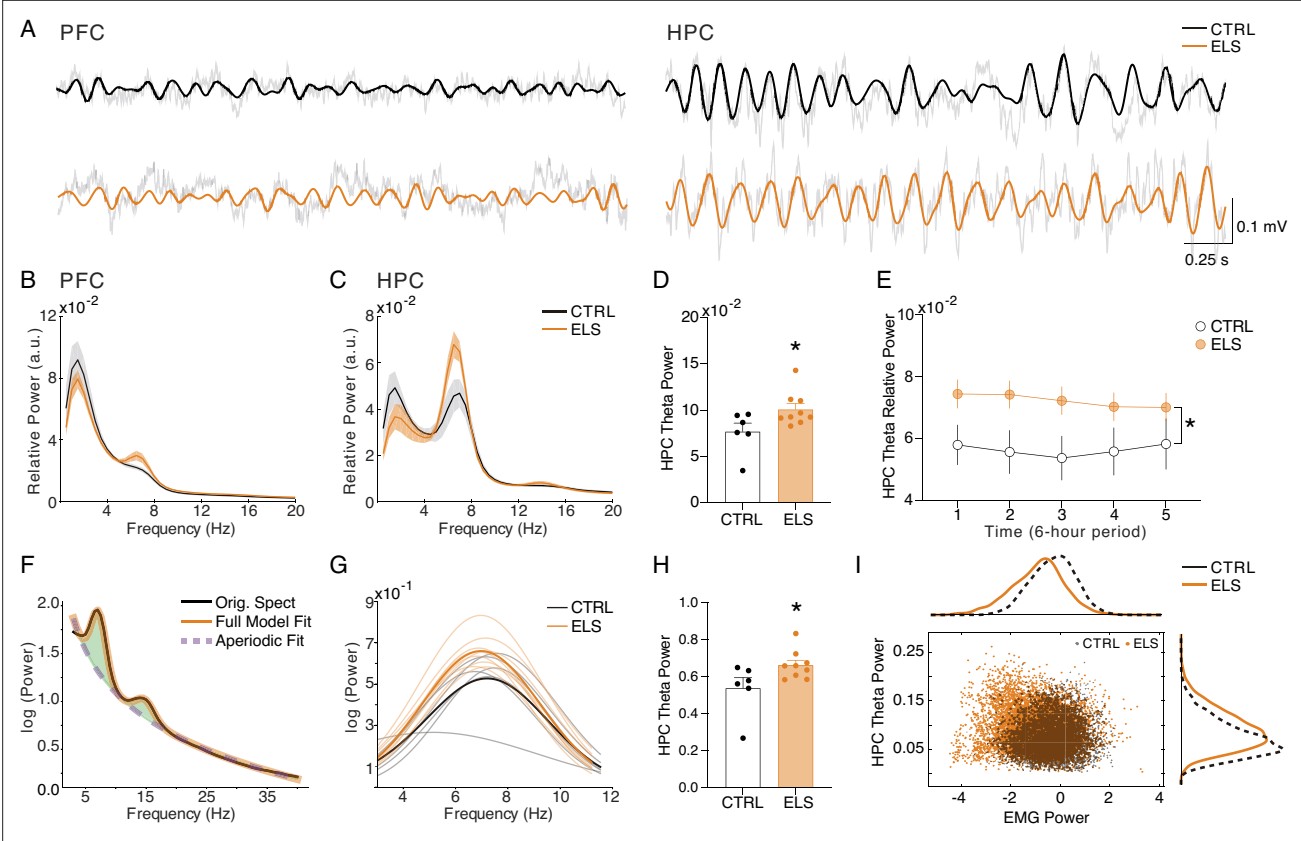

**Figure 7.** Increased HPC theta power during behavioral activity in early-life seizure (ELS) animals. (**A**) Representative raw local field potentials (LFPs) (transparent black lines) and theta-filtered signal of control (CTRL) (black) and ELS (orange) from the PFC (left) and HPC (right) during active wake (ACT). (**B**) Relative power spectrum densities (PSD) of the PFC during ACT. (**C**) Relative PSD of the HPC during ACT. (**D**) ELS rats present an enhanced HPC theta power during ACT (*t*-test). (**E**) Theta relative power during each of the 6 hr periods of recording, showing a general increase in the ELS animals that is more pronounced in the initial 18 hr of recording (linear mixed-effects model). (**F**) Parametric model used to distinguish oscillatory from aperiodic component. (**G**) Oscillatory components extracted from ACT PSDs used to calculate theta power (Mann-Whitney). (**H**) Theta power increase in ELS rats is due to a true oscillatory component and not from the aperiodic activity in ACT states. (**I**) Theta power increase in ELS rats is independent of locomotor activity. Note from the distribution of relative theta power and Z-scored EMG (trapezius) values for all rats that EMG power is not related to the regions of greater theta power in the ELS animals. *p<0.05. CTRL n=6, and ELS n=9. Error bars represent the mean ± SEM.

of time awake, number of persistent episodes in each state (Two-way RM ANOVA, $F_{(1,13)}$ = 0.6500, p>0.4346; *Figure 6E–G*), and percentage of time in each state (Two-way RM ANOVA, $F_{(1,13)}$ = 0.5425, p>0.4745; *Figure 6H*).

We also examined if ELS animals displayed ictal activity during recordings. We found that 44.45% of the ELS animals presented electrographic discharges similar to spike-wave discharges (SWD) or rhythmical spiking (*Figure 6H–I*). The mean duration of the electrographical ictal activity was 6.614±0.1416 s (coefficient of variation: 4.281%) and the total duration was 38.98±11.25 min (coefficient of variation: 57.70%). SWDs occurred mainly during NREM sleep and less often in QWK and ACT but rarely occurred during REM epochs (*Figure 6J–M*). The SWDs were not found in the CTRL rats, indicating that ELS induces an epileptogenic process.

## Increased HPC theta power during behavioral activity in ELS animals

We initially concentrated our investigation on the ACT behavioral state to associate oscillatory patterns with the behavioral abnormalities observed. We found that ELS animals presented a higher theta power during ACT states in the HPC (*t*-test, $t_{(13)}$ = 2.259, p=0.0417; *Figure 7A–D*), but not in the PFC (*t*-test, $t_{(13)}$ = 0.2636, p=0.7961; *Figure 7B*). This increased theta activity is more prominent during the first periods of recording and exploration of the new environment (linear mixed-effects model, t=2.10, p=0.055; Uncorrected *t*-tests for period comparison: $t_{(13)}$ = 2.145, 2.367, and 2.319, p=0.051,

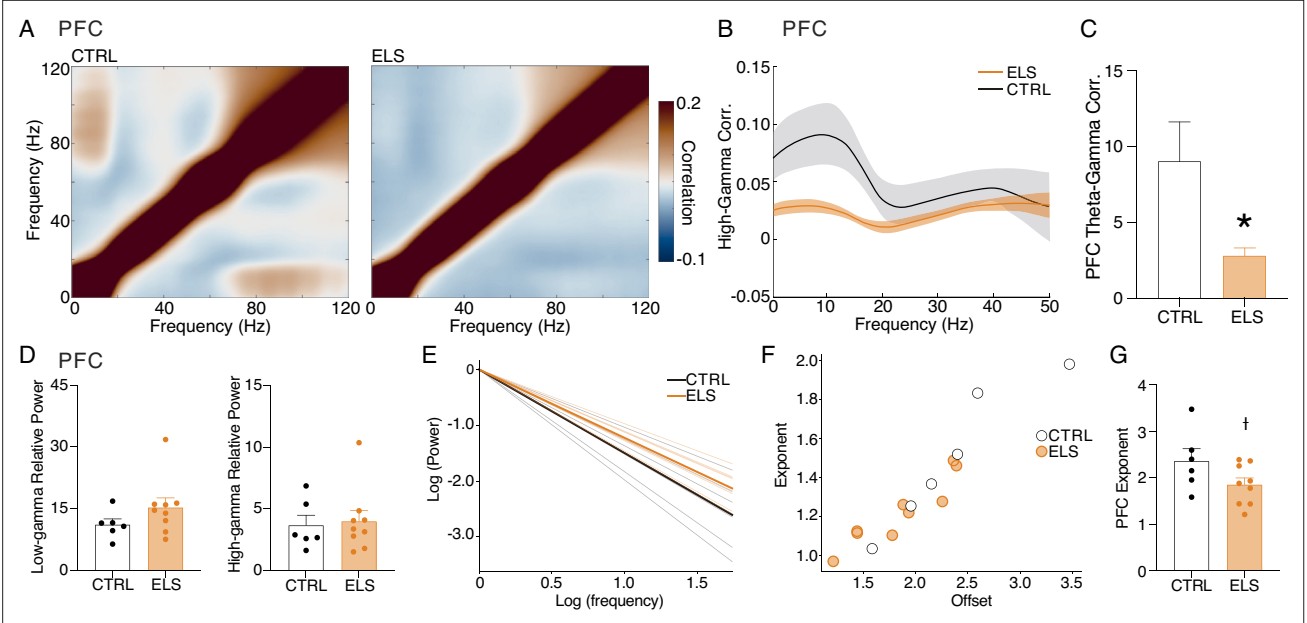

**Figure 8.** Early-life seizure (ELS) promotes abnormal gamma coordination in the PFC during behavioral activity. ELS rats present an abnormal gamma activity in the PFC. (**A**) Grand-average amplitude correlogram for control (CTRL) (left) and ELS (right) group. Note the absence of amplitude correlation between theta and gamma activity in the ELS group. (**B**) High-gamma (65–100 Hz) correlation with low-frequency activity. Note the peak around theta oscillation in the CTRL group and the absence in ELS. (**C**) ELS present an impaired theta-gamma amplitude correlation (*t*-test). (**D**) There is no difference between the low- (top) and high-gamma (bottom) average power. (**E**) Aperiodic activity extracted from the PFC of each rat during active wake (ACT) state. (**F**) Distribution of the parameters of the aperiodic component (exponent and offset) for each rat. (**G**) ELS rats show a tendency to reduction (p=0.0656, *t*-test) in the exponent parameter of the aperiodic component, as illustrated by the flatter spectra. † p<0.1, *p<0.05. CTRL n=6, and ELS n=9. Error bars represent the mean ± SEM.

0.034, and 0.037 for periods 1, 2, and 3, respectively; *Figure 7E*). To further explore whether theta power increase reflects a true oscillation and is not due to alterations in the aperiodic components of the power spectrum densities (PSD), we also performed a PSD parametrization to separate the theta oscillation (*Figure 7F*). We found an increase in the theta oscillatory activity in ELS animals during ACT states, indicating an actual oscillatory effect (Mann-Whitney, *U*=7, *n* = (6,9), p=0.0176; *Figure 7G–H*). Hippocampal theta is well known to occur during periods of locomotor activity, e.g., postural changes, head movements, and rearing (*Buzsáki and Moser, 2013*) representing a possible confounder. To control for this, we co-examined HPC theta power and electromyogram (EMG) distributions during active states in CTRL *vs.* ELS rats (*Figure 7I*). Indeed, theta oscillatory increase in ELS animals did not appear to be related to more movement, as we did not observe a significant association between EMG and theta power. In contrast, we found higher theta power in ELS animals even at low levels of EMG activity (*Figure 7I*).

## ELS promotes abnormal gamma coordination in the PFC during behavioral activity

Considering the extensively documented involvement of PFC gamma activity in cognitive and sensory processes, as well as its alterations observed in animal models of neurodevelopmental and psychiatric disorders such as ASD and schizophrenia (*Rojas and Wilson, 2014*; *Sigurdsson, 2016a*), we investigated the possible effects of ELS on PFC gamma activity during active states. ELS animals showed a reduced theta-gamma amplitude correlation during ACT states (*t*-test, $t_{(13)}$ = 2.851, p=0.0136; *Figure 8A–C*), indicating that high-gamma activity is less coordinated, despite the increase in theta power. Interestingly, there were no increases in both low or high-gamma power (low-gamma: Mann-Whitney, *U*=16, *n* = (6,9), p=0.2238; high-gamma: Mann-Whitney, *U*=24, *n* = (6,9), p=0.7756; *Figure 8D*). Amplitude correlation is an indicator of large-scale cortical interactions that govern cognition (*Siegel et al., 2012*). Thus, the lower theta-high gamma correlations found here may be related to the cognitive disruptions caused by ELS.

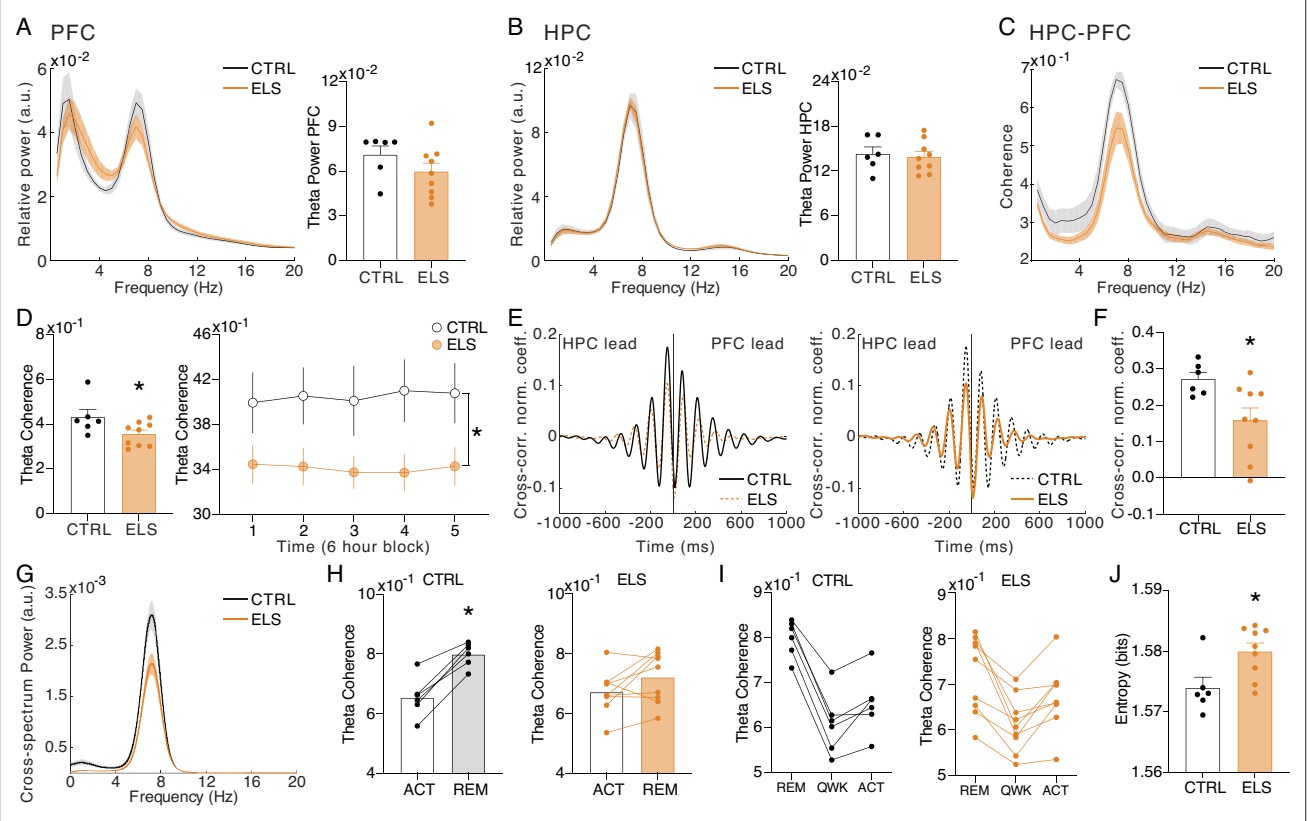

**Figure 9.** ELS impairs hippocampus-prefrontal cortex (HPC-PFC) coherence during rapid eye movement (REM) sleep. (**A**) Power spectrum densities (PSD) in the PFC during REM. (**B**) PSD in the HPC during REM. No significant alterations in theta power during REM are observed in early-life seizure (ELS) rats. (**C**) Theta coherence is diminished in ELS rats. (**D**) Significant effect on the HPC-PFC theta coherence by averaging the entire recording (left, *t*-test) and across all periods (right, linear mixed-effects model). (**E**) Time domain cross-correlation between PFC and HPC LFPs during rapid eye movement (REM) epochs highlighting control (CTRL) (left) and ELS (right) group averages. (**F**) Reduction in the peak of correlation between –80 to –20 ms in ELS rats (*t*-test). (**G**) Average cross-spectrum power density showing a reduced theta peak in the ELS rats. (**H**) Theta HPC-PFC coherence increases from active wake (ACT) to REM states. We can observe that all the CTRL rats present an increase from ACT to REM (paired *t*-test), while in ELS rats this is not observed. Paired *t*-test *p<0.05. (**I**) Theta coherence dynamics across the sleep-wake cycle. CTRL rats display a homogeneous dynamics with theta coherence being maximal at REM, minimal in quiet wake (QWK), and intermediate in ACT states. This dynamic is disrupted in ELS rats that display a higher variance between animals, with some animals displaying similar coherence in REM and ACT. (**J**) Entropy of the theta coherence dynamics is higher in ELS rats, representing the more homogeneous values across states (*t*-test). CTRL n=6, and ELS n=9.

Next, we found a reduction in the PFC spectral exponent in ELS animals assessed by aperiodic PFC activity during ACT state (note the flatter spectra of ELS in *Figure 8E*), distribution of the aperiodic component (exponent and offset, *Figure 8F*), and comparison in exponent parameter between groups (*t*-test, $t_{(13)}$ = 2.010, p=0.0656; *Figure 8G*). Reductions of the spectral exponent have been discussed to indicate increased excitation/inhibition imbalance (*Gao et al., 2017*). This abnormal gamma dynamics in the ELS indicates a less coordinated oscillatory activity, which may reflect impaired limbic-cortical or cortico-cortical interactions that underlie cognitive dysfunctions.

## ELS impairs HPC-PFC coherence during REM sleep

HPC-PFC connectivity and information flow during REM sleep are essential for memory consolidation (*Lesting et al., 2013*). Because we found cognitive impairments in ELS animals, we investigated HPC-PFC interactions during REM. First, we observed that the theta power in the PFC and HPC during REM is not different between groups (PFC: *t*-test, $t_{(13)}$ = 1.320, p=0.2096; HPC: *t*-test, $t_{(13)}$ = 0.3557, p=0.7278; *Figure 9A–B*). However, we found that ELS rats show reduced HPC-PFC connectivity in the theta band, measured by spectral coherence during REM sleep (*t*-test, $t_{(13)}$ = 2.183, p=0.0480; *Figure 9C–D*). This difference occurs throughout the entire recording, as observed by the comparisons for all 6 hr periods (linear mixed-effects model, t=–2.27, p=0.0408; *Figure 9D*, right). We further

confirmed this difference by analyzing the cross-correlation and cross-spectrum power of the HPC-PFC LFP during REM (*Figure 9E–G*). Cross-correlation analysis reveals a clear directionality between signals from the HPC to the PFC in both groups that happens markedly at a specific lag (−80 to −20 ms; *Figure 9E*), in which ELS rats present the most profound decrease in the correlation peak (*t*-test, $t_{(13)}$ = 2.533, p=0.0250; *Figure 9F–G*).

We noticed from inspecting the theta coherence changes between ACT and REM – the two states with the strongest theta coherence – that all CTRL animals exhibited greater theta coherence during REM (paired *t*-test, $t_{(5)}$ = 8.023, p=0.0005; *Figure 9H*), while in the ELS group, part of the animals did not present such an increase (paired *t*-test, $t_{(8)}$ = 1.924, p=0.09; *Figure 9H*). This dynamic is further illustrated by observing the coherence changes across REM, QWK, and ACT states. While most CTRL rats present an inclined decrease from REM to QWK and a slight increase from QWK to ACT, ELS rats display a flatter curve (*Figure 9I*). By quantifying the homogeneity of the distributions of average theta coherence values across states with entropy, we found a significantly higher entropy (more homogeneous) in ELS than CTRL (*t*-test, $t_{(13)}$ = 2.695, p=0.0184; *Figure 9J*). These results indicate that ELS promotes impairment in HPC-PFC connectivity during REM sleep and in the regulation of state-dependent dynamics.

## ELS rats display REM-like oscillatory dynamics during active behavior

The distinct neural processes occurring at each state during the sleep/wake cycle are crucial for brain homeostasis. After we noticed that ELS animals show less distinction of some electrophysiological activities between brain states, particularly REM and ACT, we decided to carry out an in-depth characterization of the long-term effects of ELS on HPC-PFC state-dependent oscillatory dynamics across the sleep-wake cycle. For that, we used the state map framework, which employs two spectral power ratios, allowing consistent characterization of all major brain states and revealing a dynamic global structure produced by the collective activity of forebrain structures (*Dzirasa et al., 2006*; *Gervasoni et al., 2004*). First, we verified that the state maps reflected other brain and muscle activities that were not directly used for the map construction, such as the concentrated distribution of the PFC delta power during NREM, of the HPC theta/delta ratios in REM, of the EMG power in awake states, and of HPC-PFC theta coherence in both REM and awake states (*Figure 10A*). Thus, as expected, our state maps presented three distinct clusters that precisely located REM, NREM, and ACT epochs (*Figure 10B*).

Using this approach, a previous study reported an overlap between awake and REM states in a hyperdopaminergic mouse model (*Dzirasa et al., 2006*). Because we found ELS-related functional changes specifically during ACT and REM, we sought to investigate the convergence between these two states. Remarkably, we also found that ELS rats showed a significant similarity between the REM and ACT states (*Figure 10B*). While in CTRL rats the distributions of Euclidean distances were larger, homogeneous, and symmetrical (*Figure 10C*), ELS rats presented highly variable patterns across animals, most of which with skewed distributions concentrated at shorter distances (*Figure 10C*). Indeed, the Euclidean distance was significantly reduced in ELS (*t*-test, $t_{(13)}$ = 2.326, p=0.0369; *Figure 10D*), suggesting that the spectral features of REM and ACT are more similar in ELS animals.

Interestingly, we observed that this difference in state similarity was stronger in the initial 12 hr of recording, which corresponds to the periods of habituation to the novel environment (*t*-test, for periods 1–2: $t_{(13)}$ = 2.45, p=0.029; for periods 3–4: $t_{(13)}$ = 0.9409, p=0.3638; *Figure 10E*).

To further demonstrate the higher similarity between REM and ACT states in ELS animals, we used machine learning algorithms in an attempt to separate these brain states through a *brute-force* approach. We observed that the two algorithms tested had a worse discrimination performance in the ELS rats, as verified by the smaller area under the receiver operating characteristic curve (AUC-ROC) (*t*-test, for support vector machine: $t_{(198)}$ = 99.59, p<0.0001; for random forest: $t_{(198)}$ = 107.6, p<0.0001; *Figure 10F*). This worse discriminative performance also arises when using random forest algorithm (which presents the best discriminative performance) on the original whole-spectrum high-dimensional variables (*t*-test, for PFC power: $t_{(198)}$ = 21.34, p<0.0001; for HPC power: $t_{(198)}$ = 58.78, p<0.0001; for coherence: $t_{(198)}$ = 80.82, p<0.0001; for all the spectral estimates combined: $t_{(198)}$ = 39.23, p<0.0001; *Figure 10G*). This lack of discriminative performance once again corroborates spectral similarities between REM and ACT states in ELS rats.

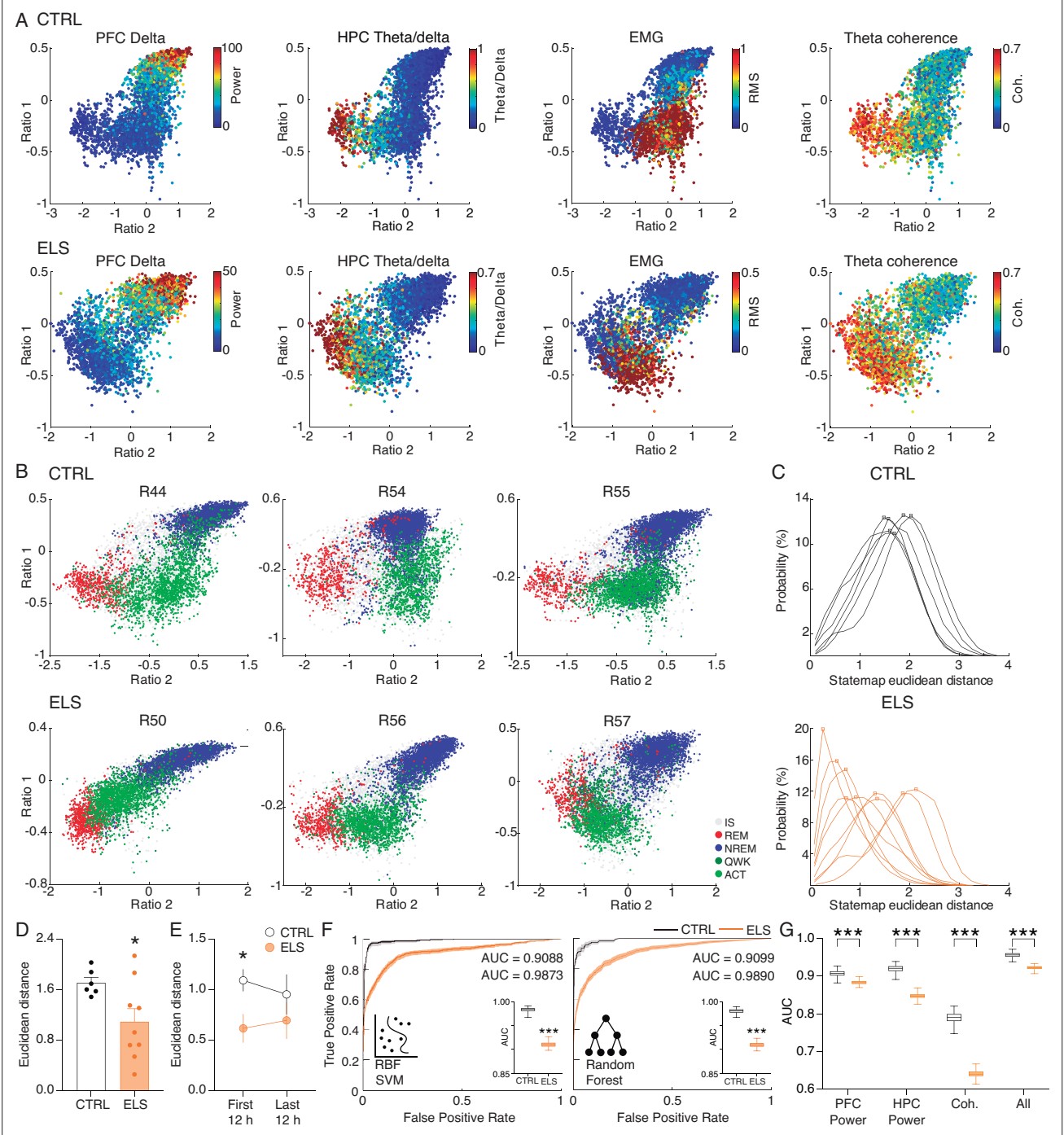

**Figure 10.** Early-life seizure (ELS) rats display rapid eye movement (REM)-like oscillatory dynamics during active behavior. (**A**) State map construction reflects the sleep-wake cycle and spectral dynamics for one representative control (CTRL) (top) and ELS (bottom) rat. PFC delta power is evident during non-REM (NREM), theta/delta ratio is maximum in REM, electromyogram (EMG) power describes active wake (ACT) states, and hippocampus-prefrontal cortex (HPC-PFC) theta coherence is stronger close to REM. Note that in the ELS rat, the coherence is also broadly high during the ACT state. (**B**) Representative state maps from three CTRL (top) and three ELS (bottom) rats. CTRL rats displayed a clear separation between REM (red), ACT (light green), and NREM (blue) clusters. ELS displayed an overlap between REM and ACT states. (**C**) Distributions of the Euclidean distances between each REM and ACT epoch for CTRL (top) and ELS (bottom) rats. ELS distributions were heterogeneous, many of which were skewed to shorter distances. (**D**) Average Euclidean distance showing that ELS rats have REM-like oscillatory dynamics during active behavior (*t*-test). (**E**) REM-like oscillations during ACT were more pronounced in ELS during the initial 12 hr of recording (*t*-test). (**F**) Machine learning approach to decode REM *versus* ACT states. We used radial basis function support vector machine (SVM) and random forest (RF) algorithm in order to classify REM and ACT epochs in CTRL (black) or ELS (orange) rats. Using a bootstrapped confidence interval, we observed that the discriminative performance was worse in ELS animals, even when using

*Figure 10 continued on next page*

'brute-force' algorithms. (**G**) A machine learning algorithm (random forest) fitted on whole-spectrum data also shows a significantly worse performance in ELS for PFC power, HPC power, coherence, and all estimates combined (*t*-test). † p<0.1, *p<0.05, **p<0.01, ***p<0.001. CTRL n=6, and ELS n=9. Error bars represent the mean ± SEM.

## ELS rats present behavioral and neurochemical sensitivity to dopaminergic activation

Similar to our findings on ELS, REM-like neural oscillations during wakefulness were described in pharmacological and genetic animal models of hyperdopaminergia (*Dzirasa et al., 2006*). Particularly, hyperdopaminergia has also been implicated in experimental models of schizophrenia as underlying impairments of sensorimotor gating and working memory deficits (*Grace, 2016*), whereas dopamine is known to optimally regulate synaptic plasticity (*Otani et al., 2015*). Taken together, these convergent findings suggest altered dopaminergic transmission as a plausible candidate to underlie some of the major effects of ELS. To investigate whether ELS promotes alterations in dopaminergic transmission or sensitivity, we conducted a final experiment assessing the behavioral and neurochemical sensitivity to dopaminergic activation using apomorphine (unspecific dopaminergic agonist). We examined behavioral sensitivity by comparing the locomotion produced by apomorphine in the open field (*Figure 11A*). Our data demonstrate that apomorphine produces a higher increase in locomotion in ELS rats when compared to CTRL across time blocks (Two-way RM ANOVA, treatment effect: $F_{(1,27)}$ = 4.260, p=0.0487, Sidak's *post-hoc:* 5 min p=0.0376; *Figure 11B*) or between average normalized distances (Mann-Whitney, *n(CTRL, ELS)*=14/15, *U*=32, p=0.0009; *Figure 11B* inset).

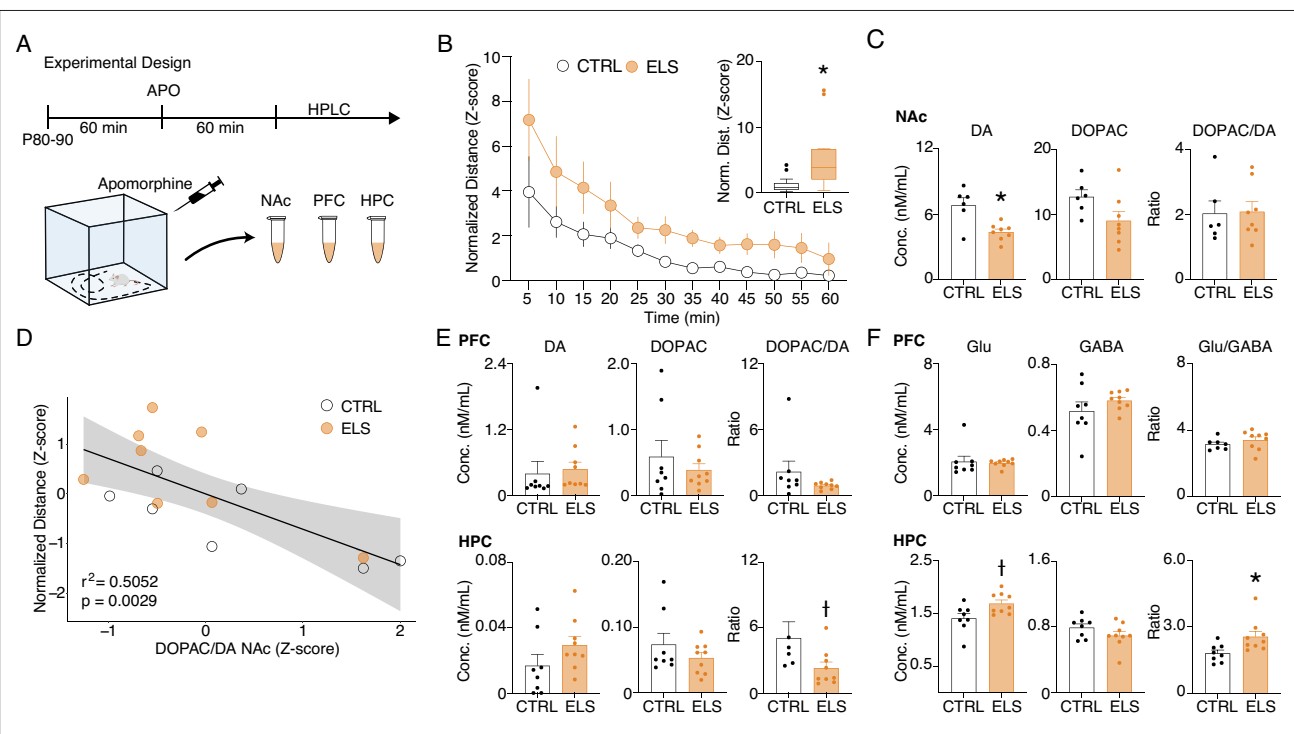

**Figure 11.** Early-life seizure (ELS) rats present behavioral and neurochemical sensitivity to dopaminergic activation. (**A**) Experimental design representing quantification of locomotion in the open field, apomorphine (APO) injection, and neurochemical analysis. (**B**) Normalized locomotion across time blocks after APO injection, demonstrating increased locomotion of ELS rats induced by APO (Two-way RM ANOVA). Average normalized locomotion after APO injection (inset, Mann-Whitney). *CTRL n=15, and ELS n=14.* (**C**) Concentration of dopamine and metabolites in the NAc. (**D**) Generalized linear model (GLM) for the total distance after APO indicating 3,4-dihydroxyphenylacetic acid (DOPAC)/dopamine (DA) ratio in the nucleus accumbens (NAc) as a strong significant variable associated with behavioral sensitivity to APO. (**E**) Concentration of dopamine and metabolites (left) in the PFC (upper panel) and HPC (lower panel). Concentration of glutamate and gamma-aminobutyric acid (GABA) (right) in the PFC (upper panel) and HPC (lower panel), using Sidak's multiple correction procedure for statistical comparison. CTRL n=8, and ELS n=9. †Uncorrected p<0.05, * multiple comparison corrected p<0.05. Error bars represent the mean ± SEM.

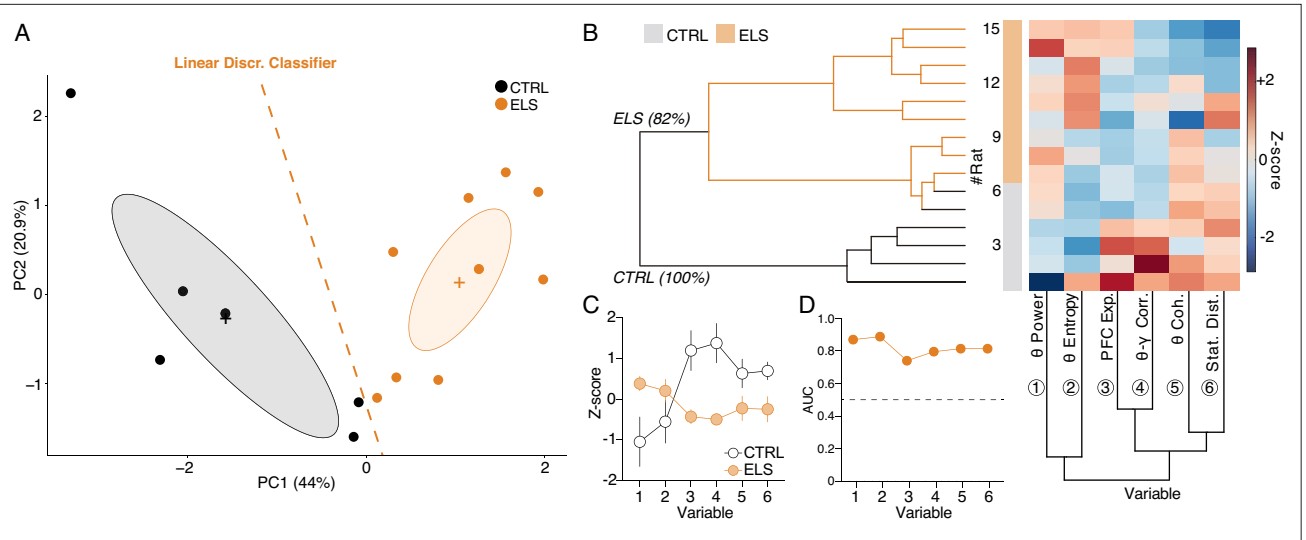

**Figure 12.** HPC and PFC oscillatory dynamics discriminate early-life seizure (ELS) rats. (**A**) A linear discriminant classifier (dotted orange line) using the first two principal components (PCs) of relevant electrophysiological variables completely distinguishes ELS from control (CTRL) rats (100% accuracy; 93.34% leave-one-out cross-validation). Note that the PC1 alone completely discriminates ELS rats. (**B**) Unsupervised hierarchical clustering using the electrophysiological variables (bottom dendrogram) almost completely dissociates ELS from CTRL rats (left dendrogram). Inside the ELS cluster, we can observe one complete ELS branch, but the other overlapping with some CTRL animals, suggesting a graduality in the pathophysiological effects of ELS. (**C**) Z-score values of each variable, showing distinct patterns of activity for each group. (**D**) All the electrophysiological variables are good discriminators of CTRL and ELS rats (AUC-ROC >0.7). CTRL n=6, and ELS n=9. Error bars represent the mean ± SEM.

Immediately after the behavioral test, we collected samples of nucleus accumbens (NAc), PFC, and HPC to quantify neurotransmitters. The NAc was included in this analysis because it is highly implicated in dopamine (DA) sensitivity and exploratory behavior (*Salamone and Correa, 2012*). Our data demonstrate that apomorphine reduces dopamine in the NAc of ELS rats, with no effects on dopamine turnover, as measured by the DOPAC/DA ratio (Sidak's multiple correction procedure, $t_{(12)}$ = 3.656, p=0.0033; *Figure 11C*). Still, a GLM model implicated NAc dopamine turnover as a significant predictor of apomorphine-induced hyperlocomotion (coefficient = –0.7108, p=0.00297; *Figure 11D*). Consistently, apomorphine is known to reduce endogenous DA release and DA metabolism in rodents and humans (*de La Fuente-Fernández et al., 2001*; *Ozaki et al., 1989*). We also observed a reduction in dopamine turnover in the HPC of ELS rats (Sidak's multiple correction procedure, $U$=9, $p$=0.0164; *Figure 11E*) but no significant effects in the PFC.

Interestingly, we found that the HPC of ELS rats shows a higher glutamate (Glu) content and higher Glu/GABA ratio than CTRL animals following apomorphine injection (Sidak's multiple correction procedure, Glu: $t_{(15)}$ = 2.566, $p$=0.0215; Glu/GABA: $U$=9, $p$=0.0079; *Figure 12F*) indicating possible disturbances in the excitation/inhibition balance.

## HPC-PFC oscillatory dynamics discriminate ELS individuals

After identifying the relevance of specific electrophysiological variables throughout the study, we sought to investigate if collective patterns of the electrophysiological covariates across individuals could assemble a multidimensional oscillatory signature that distinguishes the ELS subjects. Based on relevance, particularity, and state specificity, for this investigation, we chose: (1) HPC theta power (*Figure 7H*), (2) theta coherence entropy (*Figure 9J*), (3) PFC exponent (*Figure 8G*), (4) PFC theta-gamma correlation (*Figure 8C*), (5) theta coherence (*Figure 9D*), and (6) state map distance (*Figure 10D*). By performing PCA, we observed that the first two principal components completely discriminated ELS from CTRL animals (*Figure 12A*). This discrimination was further confirmed by fitting a linear discriminant classifier (traces on *Figure 12A*) that achieved a 100% accuracy using the entire dataset and 80% accuracy on leave-one-out cross-validation.

Furthermore, unsupervised clustering almost completely discriminated ELS from CTRL rats dividing all the animals into two major clusters: a predominantly ELS cluster (81.82% of ELS animals) and a complete CTRL cluster (100% of CTRL animals; *Figure 12B*). The hierarchical clustering approach

evidenced a progression of ELS-related oscillatory alterations, with a subcluster presenting an overlap between ELS and CTRL but the other with a more pronounced abnormal spectral fingerprint (*Figure 12B*). In addition, the values of each variable are distinct in each group, which demonstrates that the electrophysiological variables used here are jointly affected by ELS at the functional level. Finally, we show that all the single electrophysiological variables have a high discriminative value (AUC >0.7) (*Figure 12D*), yet, their multivariate linear combination represented by PC1 scores (*Figure 12A*) has the best classification performance with an AUC of 1.

## Discussion

Here, we showed that although ELS does not induce significant neuronal death, it leads to long-term cognitive impairments and behavioral abnormalities. These effects are correlated with neuronal inflammation, hypersensitive dopaminergic neurotransmission, and aberrant HPC-PFC synaptic plasticity. We also found that ELS rats exhibit brain state-dependent alterations – particularly during active behavior and REM sleep – characterized by increased theta power and impaired coordination of local and long-range activity. In addition, ELS presents an abnormal oscillatory state during active behavior oddly similar to REM sleep. Finally, the collective pattern of oscillatory alterations accurately distinguishes ELS from control animals.

### Common pathophysiological mechanisms underlying ELS consequences and psychiatric-like impairments

Remarkably, the behavioral alterations found here are well-established translational features of neurodevelopmental disorders and psychiatric disorders related to epilepsy. Studies usually describe increased locomotion (*Kubová and Mareš, 2013*) and working memory deficits (*Tsai et al., 2012*) as consequences of ELS. In addition, we also found a reduced PPI, which is less studied but represents a well-established correlate of positive symptoms in clinical psychosis and sensorimotor alterations of ASD (*Cheng et al., 2018*; *Turetsky et al., 2007*; *Ziermans et al., 2011*). Here, we demonstrated a multivariate *sensorimotor factor* comprising a collection of behaviors frequently reported in animal models of schizophrenia, ADHD, and ASD (*Ey et al., 2011*; *Regan et al., 2022*; *Sigurdsson, 2016a*). Furthermore, we demonstrated a *cognitive factor* mainly representing working memory deficits, a hallmark in several neurodevelopmental models and psychiatric comorbidities, particularly schizophrenia (*Cope et al., 2016*). Importantly, the two factors were uncorrelated, but both resulted from ELS. This lack of association between cognitive and sensorimotor impairments suggests two neurobiological substrates, which are confirmed by our multivariate analysis, indicating the association of LTP to cognitive deficits and neuroinflammation to sensorimotor alterations.

Surprisingly, we found that cognitive impairment was associated with an abnormally increased susceptibility of HPC-PFC circuits to undergo LTP. This appears to be in contrast with previous studies, which reported reduced hippocampal LTP and cognitive deficits after ELS (*Chang et al., 2003*; *O'Leary et al., 2016*). However, no hyperexcitability or spontaneous seizures were described in these studies (*O'Leary et al., 2016*). In turn, *Notenboom et al., 2010* used an animal model of febrile ELS and demonstrated increased LTP in HPC slices related to the reduction of seizure threshold, hyperexcitability, and spontaneous seizures later in life (i.e. epileptogenesis) (*Dubé et al., 2006*; *Dube et al., 2000*). Indeed, paroxysmal activity has been shown to acutely increase synaptic efficacy and hippocampal LTP, thereby reducing the threshold for seizure induction (*Ben-Ari and Gho, 1988*; *Minamiura et al., 1996*). In this context, the enhanced LTP observed in our data indicates a relationship with epileptogenesis, which concurs with the epileptiform discharges observed in ELS rats and the increases observed in the hippocampal Glu/GABA ratio.

We describe an inverted-U relationship between LTP and working memory, suggesting an optimal level of LTP susceptibility is required for adequate working memory performance. Aberrant synaptic plasticity could be a common mechanism between cognitive deficits and epileptogenesis induced by ELS. Hyperexcitable HPC-PFC circuits could facilitate the spread of epileptiform activity and promote alterations in synaptic saturation and excitatory/inhibitory balance, ultimately leading to cognitive deficits. In this sense, the abnormally increased LTP could impair sensory processes during working memory and explain the poor performance we observed in ELS rats. In a neural network with aberrantly enhanced LTP, synaptic plasticity could be biased toward potentiation, resulting in a lower

ability to retain information or adapt to the environment (*Navakkode et al., 2022*). This rationale is supported by several genetic studies, including on risk genes for neuropsychiatric disorders and autism that described relationships between enhanced LTP and impaired learning (*Garcia-Alvarez et al., 2015*; *Meng et al., 2002*; *Nuytens et al., 2013*; *Yokoi et al., 2015*). Interestingly, hyperdopaminergic signaling enhanced HPC-PFC LTP dose-dependently following an 'inverted-U' function, where too high or too low dopamine levels disrupted LTP induction, indicating that altered DA neurotransmission could underlie the LTP abnormalities in ELS rats (*Goto and Grace, 2006*; *Otani et al., 2015*).

We also report an increase in GFAP expression in ELS animals. Inflammatory processes initiated by a brain insult could lead to functional impairments and, ultimately, psychiatric symptoms (*Vezzani et al., 2013*). In ELS models, long-term glial activation has been related to behavioral impairments and epileptogenesis (*de Oliveira et al., 2008*), while treatment with inflammatory inhibitors resulted in reductions of GFAP and cognitive impairments (*Abraham et al., 2012*; *Somera-Molina et al., 2007*). High levels of pro-inflammatory substances and neuroinflammation have been documented in animal models and patients with schizophrenia and ASD (*Prata et al., 2017*; *Vezzani et al., 2013*). Noteworthy, the HPC of patients with temporal lobe epilepsy (TLE) and interictal psychosis present greater GFAP and microglia activation than TLE patients without psychosis (*Peixoto-Santos et al., 2015*). Additionally, GFAP is elevated in the frontal cortex of autistic subjects (*Laurence and Fatemi, 2005*). These findings are in accordance with clinical and preclinical biomarkers bridging psychosis and ASD to dopaminergic disturbances resulting from ELS.

In adult SE studies, behavioral abnormalities and cognitive deficits have been positively correlated to neuronal loss and spontaneous seizure frequency (*Wolf et al., 2016*). It is suggested that the etiological mechanisms of neuropsychiatric-related alterations and epileptic excitability spread progressively over time. Indeed, the behavioral impairments in ELS models tend to be more subtle than in animal models of TLE or hyperdopaminergia. It is possible that the observed alterations in neuroinflammation, neuronal dynamics, and neurochemistry represent initial epileptogenic pathological mechanisms that could progress to spontaneous seizures, progressive neuronal loss, and more intense behavioral manifestations (*Dubé et al., 2006*; *Kubová and Mareš, 2013*).

## Dysfunctional patterns of HPC-PFC oscillatory dynamics

Our results demonstrate a locomotion-unrelated increase in HPC theta power during active behavior in ELS animals. Theta oscillations underlie various cognitive processes and are believed to serve as a temporal organizer for multiple functions, such as sensorimotor integration (*Buzsáki and Moser, 2013*). However, increases in HPC theta power are also observed in hyperdopaminergic-like states, such as following the administration of ketamine or a dopaminergic agonist (*Caixeta et al., 2013*; *Thörn et al., 2022*). Increases in theta power are reported in most studies in patients with schizophrenia (*Newson and Thiagarajan, 2018*) and have been correlated with poor cognitive performance (*Curic et al., 2021*). Thus, stronger HPC theta power does not necessarily reflect an optimal cognitive state. In fact, abnormal and increased theta power has been described as a consequence of ELS in different models and related to cognitive impairment (*Kleen et al., 2011*; *Velasquez et al., 2023*).

Here, we observed lower theta-gamma correlations in the PFC of ELS animals, suggesting a disconnection between theta power and local coordination of PFC activity. Similarly, the prolonged febrile model of ELS also resulted in a decreased coordination of gamma activity in the hippocampal circuits (*Kloc et al., 2023*). In addition, we also observed a flatter PFC power spectrum in ELS animals compared to the normal tilt of CTRL, which has been discussed to indicate excitation/inhibition imbalance (*Gao et al., 2017*). These findings occurring during active behavior could explain the cognitive-behavioral alterations we observed, at least to some extent.

Desynchronization between limbic-frontal circuits is also reflected by reduced theta coherence during REM. Genetic and neurodevelopmental models of cognitive disorders exhibit a decrease in HPC-PFC theta synchrony correlated with poor spatial working memory performance and anxiety behaviors (*Cunniff et al., 2020*; *Sigurdsson, 2016a*; *Sigurdsson and Duvarci, 2016b*). Interestingly, increased HPC-PFC theta coherence was described in a model of repetitive ELS (*Holmes et al., 2015*; *Mouchati et al., 2019*). Specifically, this enhanced coherence was observed during cognitive performance and associated with a mechanism of neuronal compensation (*Kleen et al., 2011*; *Niedecker et al., 2021*). However, during REM sleep, these studies reported a decreased coherence and phase-locking value,

indicating different synchronization dynamics depending on the brain state (*Niedecker et al., 2021*). Here, we observed impaired theta coherence only during REM sleep, which is the brain state with the expected highest coherence values, indicating that ELS animals could not reach or maintain this connectivity at the same maximum capacity as CTRL animals. The results from oscillatory dynamics in the HPC-PFC circuit indicate persistent state-dependent abnormalities in the temporal organization and interaction of local circuits that can contribute to long-term behavioral alterations and cognitive deficits. Further exploration should investigate HPC-PFC coordination during specific cognitive demands, such as working memory tasks, which we have shown to be impaired following ELS.

Our results also demonstrated that during ACT behavior, frontal-limbic oscillatory dynamics of ELS rats are unusually similar to REM sleep. Importantly, this finding could not be explained by altered locomotion or sleep architecture. This aberrant active state was more pronounced in the initial 12 hr of recording, corresponding to the habituation to the novel environment. *Dzirasa et al., 2006* found similar results when recording hyperdopaminergic mice. The authors described a REM-like awake state in DAT-KO mice and wild-type mice injected with amphetamine. Similar to our results, the hyperdopaminergic wake state was marked by a significant increase in hippocampal theta oscillations. Additionally, treatment with antipsychotics reduced these REM-like hippocampal oscillations (*Dzirasa et al., 2006*). Our similar findings on a developmental model resulting in a psychotic-like phenotype support the relevance of intraindividual REM-wake similarity as a neural mechanism of psychosis. These results also advance previous studies, suggesting a shared involvement of dopamine in the impaired regulation of state-dependent global network dynamics in animal models of psychosis and epilepsy. Also interestingly, it is speculated that REM-wake similarity may represent a neural correlate of oneiric intrusion onto awake consciousness (*Dzirasa et al., 2006*), which, in the clinical context, could underlie potential phenomenological manifestations of sensorimotor disturbances of neurodevelopmental or psychiatric disorders.

In support of these dopaminergic alterations, we showed that ELS promotes apomorphine hypersensitivity and alterations in dopamine neurotransmission. Apomorphine sensitivity has been associated with reduced PPI, hyperlocomotion, and increased NAc dopamine release (*Ellenbroek and Cools, 2002*). Corroborating our data, neonatal SE promotes amphetamine sensitization and affects both dopaminergic and glutamatergic levels in prefrontal-striatal circuits (*Lin et al., 2009*). Importantly, alterations in dopamine signaling have been implicated in schizophrenia, ASD, and ADHD (*Grace, 2016*; *Kollins and Adcock, 2014*; *Kosillo and Bateup, 2021*). These results point to enhanced dopaminergic neurotransmission as a potential mechanism underlying the functional abnormalities observed in ELS rats.

Importantly, the observed alterations in mesolimbic dopaminergic activity, coupled with increased neuroinflammation in thalamic nuclei, imply broader-scale brain alterations beyond the confines of the HPC-PFC circuitry. While our findings on synaptic plasticity and oscillatory patterns suggest a robust connection between behavioral manifestations and dysfunctional communication within the HPC-PFC circuit, it is plausible that SE induces dysfunction on a broader network scale. Notably, theta oscillation and state-map examination may represent larger-scale limbic-cortical dynamics that extend beyond the HPC-PFC circuit. Consolidated evidence underscores the concept that cognitive and executive functions arise from the coordinated activity of large-scale networks (*Uhlhaas and Singer, 2012*). In this context, the disruption of functional large-scale networks may play a causal role in neuropsychiatric disorders and account for several specific dysfunctions associated with these conditions (*Uhlhaas and Singer, 2012*).

The aim of our work was to establish a comprehensive framework of ELS behavioral and neurobiological outcomes to provide a foundation for future research delving into targeted interventions. In this context, we acknowledge that most of our findings are correlational, therefore, studies investigating causal manipulations will be paramount for establishing cause-and-effect links and understanding the chain of biological events that happen in the aftermath of ELS leading to behavioral and cognitive deficits. Still, our initial data-driven approach was able to reveal nuanced relationships, such as multivariate associations and non-linear relationships, that could not have been revealed through simple univariate analysis or causally perturbing single factors at first. Based on our findings and the supporting literature, particular targets for the following investigations stand out. We speculate that facilitating HPC-PFC LTP susceptibility could boost working memory performance in CTRL animals while blocking it could improve it for ELS. Also, we hypothesize that long-term inhibition of

astrogliosis and neuroinflammation after ELS could attenuate subsequent sensorimotor impairments. Finally, we conjecture that aberrant plasticity and abnormal network dynamics could be normalized by dopaminergic inactivation in ELS animals. Thus, further studies performing causal manipulations will help us understand the mechanisms of ELS and bridge our findings with the development of new treatments for neurodevelopmental disorders.

In conclusion, we describe a comprehensive set of behavioral and neurobiological alterations following ELS that represent common patterns of dysfunctional circuit dynamics between developmental brain disturbances and multiple comorbidities associated. Although further studies should investigate causal links between neurobiological and behavioral alterations, our findings integrate a prominent theory about the pathogenesis of psychiatric symptoms that links stressful insults during development to increased HPC excitability, which would culminate in the hyperactivation of the meso-limbic system, resulting in behavioral dysfunctions (*Gomes et al., 2020*). Our results add depth to this theory by finding relationships among (1) neurodevelopmental insult, (2) neuroinflammation, (3) hyperexcitable HPC, (4) increased DA drive, (5) disrupted information flow in limbic-frontal circuits, and (6) psychiatric-relevant behaviors. Given the common pathophysiological and behavioral traits linking epileptogenesis and behavioral-cognitive comorbidities, we emphasize the experimental model's validity to investigate the neurobiological mechanisms of how an injury during development can lead to neuropsychiatric symptoms. Our study also demonstrates the potential of a multi-level strategy combining electrophysiology, behavior, immunohistochemistry, and neurochemistry in revealing integrated multidimensional mechanisms linking developmental impacts to epilepsy, cognition, and psychiatric symptoms.

## Materials and methods

### Animals

Wistar rats were housed in a colony room with a controlled temperature (22 ±2 °C) and a 12 hr light/dark cycle with free access to food and water. Litters contained up to eight animals with a maximum of five males. Rats were weaned at 21 days postnatally (P21). All procedures were performed according to the National Council for the Control of Animal Experimentation guidelines for animal research and approved by the local Committee on Ethics in the Use of Animals (Ribeirão Preto Medical School, University of São Paulo; protocol: 159/2014).

### Experimental designs

We used four cohorts (a total of 17 litters) to compare the effects of early-life *Status Epilepticus* (ELS) *versus* CTRL animals. The first cohort was used for acute electrophysiology during the induction of ELS (n=5, ELS only). Importantly, this group was utilized solely for the purpose of confirming that the SE protocol induced behavioral and electrophysiological ictal activity. This group was not subjected to any additional statistical tests or data analyses. The second cohort was used for behavioral tests (CTRL n=11, and ELS n=14), synaptic plasticity experiments (CTRL n=7, and ELS n=7), and immunohisto-chemistry (CTRL n=9, and ELS n=11). The reduction in sample size for LTP and immunohistochemistry experiments was influenced by practical challenges, including mortality during LTP surgery and issues related to tissue damage or loss during the immunohistochemical protocol. The third cohort was used to investigate HPC-PFC dynamics in freely moving rats (CTRL n=6, and ELS n=9). Finally, in the fourth cohort, we assessed behavioral sensitization to psychostimulants (CTRL n=15, and ELS n=14) and neurochemical quantification (CTRL n=8, and ELS n=9). The reduced sample size for neurotransmitter analysis was a deliberate selection of a subsample, considering the complexity and cost associated with the neurotransmitter quantification protocol. Rats were randomly selected for each group.

### Early-life status epilepticus

At P11-P12 we induced a 2 hr SE using lithium chloride (LiCl, 127 mg/kg, i.p., Sigma-Aldrich, USA) 18–20 hr prior to injection of methylscopolamine (1 mg/kg, i.p., Sigma-Aldrich) and pilocarpine (80 mg/kg, s.c., Sigma-Aldrich, USA) (*Kubová and Mareš, 2013*). Behavioral seizures were monitored following the Racine scale (*Racine, 1971*). SE was interrupted by diazepam (5 mg/kg, i.p., Teuto, Brazil) at the 2 hr mark. During the recovery period (1 week post-ELS) we provided food supplementation and subcutaneous rehydration with saline and monitored body temperature and weight.

## Behavior

The radial-arm maze working memory task was based on *Floresco et al., 1997*. Briefly, rats were kept at 85–90% of their initial body weight throughout the 21 days of the experiment. Training consisted of single daily sessions, each one divided into the training phase and test phase, 30 min apart. During the training session, four doors were kept open providing access to food pellets. In the test session, all doors remained open, but with pellets available only in the four arms that were closed in the previous training session. The sessions were terminated upon full pellet consumption or after 5 min. Two types of errors were quantified: Reference error: reentering arms previously visited during the training session; Working memory error: reentering an arm already visited during the test session. The sum of both categories denoted the total errors for each animal.

Three days after returning to ad libitum diet, locomotor activity was evaluated for 30 min in the open field test (*Wolf et al., 2016*).

Immediately after the open field test, rats were examined for PPI of the acoustic startle. The PPI test consisted of a habituation period (5 min) followed by 72 trials (intertrial interval: 15±8 s) with prepulse alone (PP), startling pulse alone (P), prepulse followed by a startling pulse (PP +P), and no stimulus, all of which accompanied by background noise (65 dB white noise). Startle-eliciting stimuli were presented at 120 dB for 40 ms (P) and prepulse stimuli were presented for 20 ms (PP, 1000 Hz; 71, 77, and 83 dB; 20 ms). Trials were presented in a random order, with each trial type being presented six times. We used the equation: %PPI = 100–100 × (mean startle amplitude at PP +P)/(mean startle amplitude at P) (*Wolf et al., 2016*).

In order to measure the effects of ELS on sensitization to psychostimulants, we performed the apomorphine-induced locomotion test. At P80-90, rats were placed in the open field apparatus and were allowed to roam for 60 min. Then, we injected apomorphine (APO, 1.5 mg/kg, i.p., Sigma-Aldrich, USA) and measured locomotion for another 60 min. The distance across blocks was normalized by Z-score to the mean of baseline blocks.

## Synaptic plasticity

HPC-PFC synaptic plasticity studies were performed as described in previous works from our group (*Bueno-Junior et al., 2018*; *Ruggiero et al., 2018*). Briefly, rats were anesthetized with urethane (1.2 mg/Kg in NaCl 0.15 M, i.p.) for electrode implantation. A recording electrode (Teflon-coated tungsten wires 60 μm, AM-Systems) was inserted in the medial PFC (prelimbic region (PL), anterior-posterior: 3.0 mm; medial-lateral, ML: 0.5 mm; dorsal-ventral, DV: 3.2 mm), and a bipolar stimulation electrode (twisted wires; ~500 μm inter-pole distance) was inserted in the ipsilateral HPC (intermediate CA1 AP: −5.6 mm; ML: 4.5 mm; DV: 2.5 mm). An additional burr hole was drilled over the right parietal cortex for a reference screw. Temperature was kept constant at 37±0.5°C during the experiment.

HPC was stimulated with single monophasic pulses (200 μs, 0.05 Hz, 150–300 μA, 80 ms inter-pulse interval) while recording PFC field postsynaptic potentials (fPSPs). Stimulation intensity (60–500 μA) was calibrated to evoke 50–60% of the maximal fPSP amplitude. Field recordings were conditioned through a preamplifier (100×gain, 0.03–3 kHz bandpass; Grass) and digitized at 10 kHz (ADInstruments). For LTP induction, we used a high-frequency stimulation (HFS) protocol consisting of 2 series (10 min apart) of 10 trains (0.1 Hz), each train with 50 pulses at 250 Hz (*Ruggiero et al., 2018*). fPSP amplitudes and slopes were computed as the difference and slope between the positive and negative peaks, respectively. Short-term synaptic plasticity was estimated by paired-pulse facilitation (PPF), calculated as the ratio of the fPSP2 and fPSP1 (*Lopes-Aguiar et al., 2020*). Amplitudes and slopes were normalized as ratios of the baseline mean and averaged into 10 min bins.

## Brain tissue processing

### Immunohistochemistry

We employed a standardized immunohistochemistry protocol (*Peixoto-Santos et al., 2015*) to estimate NeuN+ neuronal density and to evaluate parvalbumin-expressing (PV) interneurons, mGluR5 expression, and glial fibrillary acid protein (GFAP) reaction. Briefly, endogenous peroxidases were blocked and the antigens were exposed by microwave-induced retrieval. We used the primary antibodies against NeuN (Chemicon, USA; 1:1000), PV (Calbiochem, USA; 1:500), mGluR5 (Millipore, USA; 1:200), and GFAP (Dako, Denmark; 1:500) proteins, and biotinylated secondary antibodies

(Dako, Denmark; 1:200). Revelation was performed with Elite ABC Kit (Vector) and diaminobenzidine solution (Pierce).

Images from the regions of interest were captured at 200x magnification in the Axio Scope.A1 microscope system (Carl Zeiss). The regions of interest included the hippocampus (CA1 and GL, granule layer of dentate gyrus), entorhinal cortex (EC), infralimbic (IL) and prelimbic cortex (PL), and thalamic reticular nucleus (TRN). Images were then processed in Image J software (NIH, USA; 1.48 v). Data are shown as the density of NeuN positive cells and immunopositive area for PV, mGluR5, and GFAP.

## High-performance liquid chromatography (HPLC)

Samples were taken bilaterally from NAc, PFC, and HPC. The samples were homogenized with ultrasound in a 0.1 mol/L $HClO_4$ 0.02% $Na_2S_2O_2$ solution containing dihydroxybenzylamine (DHBA, 146.5 ng/mL), and homoserine (HSER, 10 µg/mL) as internal standards and at a dilution of 15 µL per sample milligram. We employed Shimadzu LC-10AD isocratic systems with a fluorescence detector for glutamate (Glu) and gamma-aminobutyric acid (GABA), and an electrochemical detector for dopamine (DA) and its metabolites 3,4-dihydroxyphenylacetic acid (DOPAC) as previously published (*de Castro-Neto et al., 2013*). The neurotransmitter levels were considered to reflect neurotransmitter stock in synaptic vesicles, their metabolites reflected the release, and metabolite/neurotransmitter ratios were taken as an index of neurotransmitter turnover. Standards containing all amino acids and monoamines/metabolites of interest were employed to define their respective peaks in the samples.

## Freely moving electrophysiology

### Electrophysiological recordings

Chronic head caps consisted of two eight-channel connectors (Omnetics), one for the PFC (PL, 7 wires), and another for the HPC (intermediate CA1, 7 wires) (*Bueno-Junior et al., 2017*; *Marques et al., 2022*). We additionally implanted a recording wire into the trapezius muscle to serve as an electromyogram (EMG). The EMG wire was soldered to the HPC connector. Microscrews were fastened into the skull, including a ground reference in the intraparietal area. Animals were allowed to recover for 8–9 days before recordings.

To record electrophysiological signals through sleep-wake states, rats were placed into a shuttle box apparatus inside a soundproof box with food and water ad libitum. Recordings varied between 30–48 hr, divided later into periods of 6 hr for analysis. Local field potentials (LFP) were recorded through a multichannel acquisition processor (Plexon) (1000×gain, 0.7–500 Hz bandpass filter, 1 kHz sampling rate).

### Classification of sleep-wake states

Classification of the sleep-wake cycle states was carried out through a semi-supervised machine-learning pipeline. The HPC theta (4.5–12 Hz), PFC delta (0.7–4 Hz), and EMG (100–200 Hz) power were computed in 5 s epochs without overlaps (four tapers; time-bandwidth product: 2.5). A support vector machine (SVM) (radial basis function) was trained to discriminate the epochs into: (1) awake or sleep epochs using EMG power; (2) REM or NREM epochs based on theta/delta ratio; (3) quiet wake (QWK) or active (ACT) epochs based on theta/delta ratio. Later, we classified our recordings based on 30 s epochs by grouping sets of six 5 s epochs and considering the most representative state.

### Spike wave-discharge detection (SWD)

For spike wave-discharge detection, we used the PFC recordings. First, the short-time Fourier transform (STFT) was computed within 3–70 Hz throughout 250 ms windows with 50% overlap. A detection threshold was defined as the mean +1.5 standard deviation of the absolute value of the STFT of the frequency bands 4.4–16.4, 17.6–32.8, and 35.1–55 Hz throughout the entire recording. SWD events were detected when the average STFT of the aforementioned frequency bands at one epoch exceeded the threshold. Consecutive epochs above the threshold were grouped as the same event. The beginning and the end of an event were defined as the closest epochs when the average STFT was lower than the mean +1 standard deviation of STFT of the frequency bands of interest. Lastly, the detected putative events were manually sorted by an expert.

## Spectral estimates

We divided 30 s epochs into 3 s epochs for spectral analyses. PSD and magnitude-squared coherence were obtained using Welch's method with 1 s segments, 50% overlap, and Hamming windowing. Relative power was obtained by dividing values by the sum within 0.5–55 Hz for each epoch. We defined the frequency bands: delta (1–4 Hz), theta (4.5–11 Hz), low gamma (30–55 Hz), and high-gamma (65–100 Hz) to compare estimates and obtained the spectral peak of each epoch using MATLAB function *findpeaks*.

To quantify theta oscillation power we used the FOOOF algorithm (**Donoghue et al., 2020**), which parametrizes the PSD separating the periodic and aperiodic components of the signal. We used frequencies within 3–40 Hz, the 'fixed' aperiodic mode, a minimum peak height of 0.05, and a maximum number of peaks of 4 (https://fooof-tools.github.io/fooof/). EMG data were low-cut filtered (>30 Hz) and the sum of power within 30–200 Hz was obtained.

As a synchrony measure, we also computed cross-correlations between each pair of LFPs (low-pass filter of 50 Hz) recorded from the HPC and PFC using a maximum lag of 1 s. Coefficients were normalized so that the autocorrelations at zero lag were one. We observed that the highest coefficients occurred in the range of –80 to –20 ms (HPC leading the PFC), so we obtained the maximum coefficient within this range for each rat average for comparisons. Cross-spectrum power was calculated using Welch's method for the normalized correlation coefficients from –1–1 s.

For cross-frequency amplitude spectrum correlations, we computed the absolute STFT for each 3 s epoch throughout 100 ms segments with 50% overlap and Hamming windowing across 0.5 Hz frequency bins within 0.5–200 Hz. We calculated the correlation matrix for each frequency pair using Pearson's R.

## State maps

After state classification and spectral analysis, we computed two-dimensional state maps to investigate the natural distinction between sleep-wake states. The state maps were computed according to **Gervasoni et al., 2004** and **Dzirasa et al., 2006**. First, we obtained the power *ratio 1* (4.5-9/4.5–50 Hz) and *ratio 2* (2-20/2-50 Hz) for each region and the first principal component, estimated by a singular value decomposition algorithm considering both regions, for each ratio. Finally, we applied a Hann window of 10 epochs on scores throughout time for smoothing.

To identify the intermediate states (IS), we first estimated the trajectory *speed* of epochs throughout state map dimensions as the Euclidean distance between each epoch and its predecessor. Then, we defined a threshold of 0.1 and labeled the epochs with greater speed as IS. This threshold was determined heuristically, corresponding closely to the expected value plus a standard deviation from a Poisson distribution across rats.

From visual inspection and the literature (**Dzirasa et al., 2006**; **Gervasoni et al., 2004**), the state maps usually separate three well-defined clusters heuristically corresponding to awake, REM, and NREM sleep. To assure the correspondence between state map clusters and the classification outcomes, we performed Gaussian mixture models clustering of the state maps for hree clusters and 80% posterior probabilities and computed Chi-squared tests with the classified NREM, REM, and awake (QWK +ACT) labels. Three animals were excluded for having outlier lower Chi-values. Although QWK and ACT states are often evenly distributed inside the awake cluster, QWK epochs may overlap with intermediate states closer to NREM, so the ACT epochs were considered to represent more precisely the localization of neural dynamics associated with behavior in the state map. The dissimilarity between ACT and REM states was estimated by the average pairwise state map Euclidean distance of all epochs between the two states.

We used machine learning algorithms to decode sleep-wake states from the state map. We used the ACT and REM state map *ratios 1* and *2* values. We applied a linear and radial basis function support vector machine (SVM) *fictcsvm* function in MATLAB ('KernelFunction,' 'linear,' or 'rbf') as well as a Random Forest algorithm using the *fitcensemble* function in MATLAB ('Method,' 'bag,' 'NumLearningCycles,' 50, for Random Forest) to decode states. We used the ACT and REM epochs from all the rats creating one single model for the CTRL or ELS condition. Using the same approach, we also constructed a random forest decoder model with non-reduced variables using 1–55 Hz (0.5 Hz bins) for the PFC and HPC power and coherence. Discrimination was evaluated using the area under the receiver operating characteristic curve (AUC-ROC). The pointwise confidence interval was calculated

using a 1000-repetition bootstrap. Average AUC-ROC values were calculated using the prediction of the trained model on 20% of the epochs as test data and repeating the process 100 times.

## Statistical analysis

Data were analyzed using GraphPad Prism Software version 9.00 (GraphPad Software, LLC), and R studio version 2021. We used paired Student *t*-tests for within-group comparisons and two-sample *t*-tests for between-group comparisons. Two-way ANOVA with repeated measures (RM) was performed for comparisons across time bins, using Sidak's multiple comparison tests as a *post-hoc*. To examine linear model assumptions, we analyzed the distribution of the residuals using a Q-Q plot, the Kolmogorov-Smirnov test, homoscedasticity, and the residual plot. We used the Mann-Whitney test to compare two groups and the Wilcoxon signed-rank test to paired comparisons as non-parametric alternatives. We calculated Pearson's linear correlation and Spearman's rank correlation as a nonparametric equivalent. To compare correlation and spectral coherence estimates we applied Fisher's Z transform to the data. For multiple comparisons in the neurochemical quantification, we performed Holm's procedure for multiple comparison correction across neurotransmitters and metabolites per region.

Data are expressed as the mean ± SEM. The significance level was set to 0.05 unless stated otherwise. For some multiple comparisons, statistical values are expressed as the range. Significances are expressed as $^{†}p<0.1$, $^{*}p<0.05$, $^{**}p<0.01$, and $^{***}p<0.001$.

## Linear mixed-effects models

We also constructed statistical inferential models across time controlling for individual variability and litter using linear mixed-effects models. We compared power and coherence estimates throughout time by dividing the electrophysiological recording into five periods (6 hr). Group and time (periods) were assigned as fixed effects and subject, litter, and time were assigned as random effects. We used this approach to construct a maximal random effects structure model as preconized in the literature as more generalizable (*Barr et al., 2013*). The model was estimated by restricted maximum likelihood using the *lme* function of the *nlme* package (*Pinheiro et al., 2023*) with an unstructured covariance matrix.

## Multivariate analysis

Each variable was Z-scored before multivariate analysis. We constructed correlograms to assess the relationship within behavioral and neurobiological measures. Variables were sorted by agglomerative hierarchical clustering using Ward's method and correlation as the metric. We also performed agglomerative hierarchical clustering of subjects using Ward's method and Euclidean distance as the metric to assess how unsupervised clustering by our multivariate data might match the experimental groups. An optimal number of clusters was estimated using the silhouette method and final definition was based on parsimonious interpretability of the results.

PCA, using singular-value decomposition, was used for dimensionality reduction to find collective patterns of variance within behavioral variables (behavior factors) and neurobiological variables (neurobiology factors), and to reduce each immunohistochemical measure across all brain regions into single scores (*Johnson and Wichern, 2007*; *Marques et al., 2023*; *Marques et al., 2022*). Principal components (PCs) were projected onto data and the mean score was compared between conditions.

CCA was performed to describe multivariate relationships between behavioral and neurobiological measures (*Drysdale et al., 2017*). Missing data were imputed using the multiple imputation by chained equations algorithm. For CCA, we used the first PC (PC1) and second PC (PC2) of behavioral variables as inputs for the behavioral factors and the PC1 of each immunohistochemical variable (NeuN, PV, mGluR5, and GFAP,) across all brain regions and the Z-scored LTP data as inputs for the neurobiological factors. Variables were narrowed down from 30 to 7, which is statistically appropriate for our sample size. We used the CCA package in R to obtain the canonical dimensions and performed the Hotelling-Lawley trace statistic to assess the significance of each dimension.

We used multivariate generalized linear models (GLM) with a Gaussian family and an identity link function. Quasi-Poisson regression (quasi-Poisson family and logarithmic link function) were used for count data with overdispersion (Errors in the radial maze) using a quadratic covariate (LTP). We

constructed two types of models: (1) a full model considering all the covariates; and (2) a restricted model using backward stepwise model selection based on the Akaike information criterion.

To assess the effectiveness of neurophysiological features in discriminating CTRL from ELS rats, we fitted a regularized linear discriminant classifier model on the PCA scores of selected electrophysiological variables. We used the MATLAB function *fitcdiscr* without hyperparameter optimization (*Johnson and Wichern, 2007*; *Marques et al., 2023*; *Marques et al., 2022*). The linear classifier using two predictors allowed the graphical representation and avoided the overfitting of a high number of variables. Accuracy was determined as the probability of correct category assignments using leave-one-out cross-validation. We also determined the contribution of each original neurophysiological variable by assessing their respective AUCs using a univariate logistic regression model.

## Acknowledgements

This research was funded by Fundação de Amparo à Pesquisa do Estado de São Paulo - FAPESP (RNR: 2018/02303–4; MTR: 2020/01510–6; DBM: 2022/16812–3; JPL: 2016/17882–4), Coordenação de Aperfeiçoamento de Pessoal de Nível Superior - CAPES - Finance Code 001 (DBM: 88882.328283/2019–01), and Conselho Nacional de Desenvolvimento Científico e Tecnológico - CNPq (JPL: 305104/2020–9 and 422911/2021–6; TP: 164165/2018–5). We thank Renato Meirelles e Silva, Renata Caldo Scandiuzzi and Daniela Ribeiro for technical support. We thank Ana Carolina Medeiros and Rafael D'Agosta for helping with the photomicrography and histology. We also thank Patrick Forcelli, Claudio Queiroz, Marcelo Caetano, Tonicarlo Velasco, Ricardo Saute, and Norberto Garcia Cairasco for discussions.

## Additional information

### Funding

| Funder | Grant reference number | Author |
| --- | --- | --- |
| Fundação de Amparo à Pesquisa do Estado de São Paulo | 2018/02303-4 | Rafael Naime Ruggiero |
| Fundação de Amparo à Pesquisa do Estado de São Paulo | 2020/01510-6 | Matheus Teixeira Rossignoli |
| Fundação de Amparo à Pesquisa do Estado de São Paulo | 2022/16812-3 | Danilo Benette Marques |
| Fundação de Amparo à Pesquisa do Estado de São Paulo | 2016/17882-4 | Joao Pereira Leite |
| Coordenação de Aperfeiçoamento de Pessoal de Nível Superior | 88882.328283/2019-01 | Danilo Benette Marques |
| Conselho Nacional de Desenvolvimento Científico e Tecnológico | 305104/2020-9 | Joao Pereira Leite |
| Conselho Nacional de Desenvolvimento Científico e Tecnológico | 422911/2021-6 | Joao Pereira Leite |
| Conselho Nacional de Desenvolvimento Científico e Tecnológico | 164165/2018-5 | Tamiris Prizon |

The funders had no role in study design, data collection and interpretation, or the decision to submit the work for publication.

## Author contributions

Rafael Naime Ruggiero, Conceptualization, Data curation, Software, Formal analysis, Supervision, Validation, Investigation, Visualization, Methodology, Writing – original draft, Project administration, Writing – review and editing; Danilo Benette Marques, Data curation, Formal analysis, Validation, Writing – original draft, Writing – review and editing; Matheus Teixeira Rossignoli, Data curation, Validation, Visualization, Methodology, Writing – original draft, Writing – review and editing; Jana Batista De Ross, Data curation, Formal analysis, Methodology; Tamiris Prizon, Data curation, Investigation, Methodology; Ikaro Jesus Silva Beraldo, Formal analysis; Lezio Soares Bueno-Junior, Validation, Methodology; Ludmyla Kandratavicius, Jose Eduardo Peixoto-Santos, Formal analysis, Methodology; Cleiton Lopes-Aguiar, Funding acquisition, Investigation, Methodology; Joao Pereira Leite, Conceptualization, Resources, Supervision, Project administration

## Author ORCIDs

Rafael Naime Ruggiero (ID) http://orcid.org/0000-0002-4712-6853
Danilo Benette Marques (ID) http://orcid.org/0000-0002-8293-8822
Matheus Teixeira Rossignoli (ID) http://orcid.org/0000-0002-0522-2337
Jana Batista De Ross (ID) http://orcid.org/0000-0001-6618-9166
Tamiris Prizon (ID) http://orcid.org/0000-0002-1098-546X
Ludmyla Kandratavicius (ID) http://orcid.org/0000-0003-4387-5977
Jose Eduardo Peixoto-Santos (ID) http://orcid.org/0000-0001-7461-1902
Cleiton Lopes-Aguiar (ID) http://orcid.org/0000-0001-9310-6338
Joao Pereira Leite (ID) http://orcid.org/0000-0003-0558-3519

## Ethics

All procedures were performed according to the National Council for the Control of Animal Experimentation guidelines for animal research and approved by the local Committee on Ethics in the Use of Animals (Ribeirão Preto Medical School, University of São Paulo; protocol: 159/2014).

Reviewer #1 (Public review): https://doi.org/10.7554/eLife.90997.3.sa1
Author response https://doi.org/10.7554/eLife.90997.3.sa2

---

# Additional files

## Supplementary files

• MDAR checklist

## Data availability

All behavioral, electrophysiological, immunohistochemical, and neurochemical data have been deposited at https://osf.io/3yw65/ and are publicly available. The code for the analyses presented in this paper is openly accessible at https://github.com/rafaruggiero/Ruggiero-eLife2024 (copy archived at *Rafaruggiero, 2024*).

The following dataset was generated:

| Author(s) | Year | Dataset title | Dataset URL | Database and Identifier |
|---|---|---|---|---|
| Ruggiero RN | 2024 | Dysfunctional Hippocampal-Prefrontal Network Underlies a Multidimensional Neuropsychiatric Phenotype following Early-Life Seizure | https://doi.org/10.17605/OSF.IO/3YW65 | Open Science Framework, 10.17605/OSF.IO/3YW65 |

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
