## [Editor Report · eLife assessment]

This **important** study assesses anatomical, behavioral, physiological, and neurochemical effects of early-life seizures in rats, describing a striking astrogliosis and deficits in cognition and electrophysiological parameters. The **solid** results come from a wide range of convergent techniques that were used to understand the effects of early-life seizures on behavior as well as hippocampal prefrontal cortical dynamics. This paper will be of interest to neurobiologists, epileptologists, and behavioral scientists.

---

## [Referee Report · Reviewer #1 (Public review)]

Summary:

In this paper, Ruggiero, Leite and colleagues assess the effects of early life seizures on a large number of anatomical, physiological, behavioral and neurochemical measures. They find that prolonged early life seizures do not lead to obvious cell loss, but lead to astrogliosis, working memory deficits on the radial arm maze, increased startle response, decreased paired pulse inhibition, and increased hippocampal-PFC LTP. There was a U-shaped relationship between LTP and cognitive deficits. There is increased theta power during the awake state in ELS animals but reduced PFC theta-gamma coupling and reduced theta HPC-PFC coherence. Theta coherence seems to be similar in ACT and REM states in ELS animals while in decreases in active relative REM in controls.

Strengths:

The main strength of the paper is the number of convergent techniques used to understand how hippocampal PFC neural dynamics and behavior change after early life seizures. The sheer scale, breadth and reach of the experiments are praiseworthy. It is clear that the paper is a major contribution to the field as far as understanding the impact of early life seizures. The LTP findings are robust and provide an important avenue for future study. The experiments are performed carefully and the analysis is appropriate. The paper is well-written and the figures are clear.

Weaknesses:

The main weakness of the paper remains the lack of causal manipulations to determine whether prevention or augmentation of any of the findings have any impact on behavior or cognition. Alternatively, if other manipulations would enhance working memory in ELS animals, it would have been interesting to see the effects on any of these parameters measured in the paper. The authors now discuss the lack of causal manipulations in the discussion but have not performed new experiments to address this weakness. Also, I find the sections where correlations and dimensionality reduction techniques are used to compare all possible variables to each other less compelling than the rest of the paper (with the exception of the findings of U shaped relationship of cognition to LTP). In fact, I think these sections take away from the impact of the actual findings. The rationale for the apomorphine experiments are now explained more fully.

---

## [Author Response]

The following is the authors’ response to the original reviews.

**eLife assessment**
This important study assesses anatomical, behavioral, physiological, and neurochemical effects of early-life seizures in rats, describing a striking astrogliosis and deficits in cognition and electrophysiological parameters. The convincing aspects of the paper are the wide range of convergent techniques used to understand the effects of early-life seizures on behavior as well as hippocampal prefrontal cortical dynamics. While reviewers thought that the scope was impressive, there was criticism of the statistical robustness and number of animals used per study arm, as well as the lack of causal manipulations to determine cause-and-effect relationships. This paper will be of interest to neurobiologists, epileptologists, and behavioral scientists.

We thank Joseph Gleeson as the Reviewing Editor and Laura Colgin as the Senior Editor for considering this revision of our manuscript for publication in eLife. We appreciate the positive acknowledgment of the study and the critical points raised by the reviewers. We have addressed all the excellent comments of the two reviewers, providing a detailed response for each comment. We believe that these revisions have significantly improved the quality and rigor of our study.

We want to assure you that our experimental design was meticulously crafted, incorporating adequate control groups, and is grounded in prominent studies in systems neurophysiology focusing into early-life seizures effects, especially for capturing mild effects. We conducted statistical tests adhering to established norms and recommendations, ensuring a thorough and transparent description of the employed statistical methods. We welcome any specific suggestions to further improve this aspect.

In fact, the concerns raised by the reviewers regarding statistical robustness may stem from a misunderstanding of the rat cohorts used in each experiment. Criticism was directed at the use of only 5 animals without a control group for acute electrophysiological recording. It is essential to clarify that this group served the sole purpose of confirming that the injection of lithium-pilocarpine would induce both behavioral and electrographic seizures. Importantly, this was a descriptive result, and no statistical test or further analysis was conducted with these data. In the revised manuscript, we have made adjustments to this description, aiming to eliminate any ambiguity, particularly addressing the issue of sample size in each experiment.

Regarding the lack of causal manipulations, we fully agree that this approach would provide a deeper mechanistic understanding of our findings and is an essential next step. Still, developmental brain disturbances are linked to manifold intricate outcomes, so an initial observational exploration would offer insights about particular and nuanced relationships for following studies aimed at targeted interventions. In this context, our objective was to provide a comprehensive characterization of ELS effects to serve as a foundation for future research. While recognizing the relevance of causal manipulations, only a more sophisticated data analyses were able to reveal more complex aspects like specific multivariate associations and non-linear relationships that would not have been revealed by causally perturbing one or another factor at first. In the revised manuscript, we emphasized the limitation of lacking causal manipulations as well as the advantages of our approach. Also, we mentioned some possible targets for following perturbational investigations based on our findings.

For a more detailed discussion on these matters, we invite you to review our response to reviewers.

**Reviewer 1**
In this paper, Ruggiero, Leite, and colleagues assess the effects of early-life seizures on a large number of anatomical, physiological, behavioral, and neurochemical measures. They find that prolonged early-life seizures do not lead to obvious cell loss, but lead to astrogliosis, working memory deficits on the radial arm maze, increased startle response, decreased paired pulse inhibition, and increased hippocampal-PFC LTP. There was a U-shape relationship between LTP and cognitive deficits. There is increased theta power during the awake state in ELS animals but reduced PFC theta-gamma coupling and reduced theta HPC-PFC coherence. Theta coherence seems to be similar in ACT and REM states in ELS animals while in decreases in active relative REM in controls.Strengths:The main strength of the paper is the number of convergent techniques used to understand how hippocampal PFC neural dynamics and behavior change after early-life seizures. The sheer scale, breadth, and reach of the experiments are praiseworthy. It is clear that the paper is a major contribution to the field as far as understanding the impact of early-life seizures. The LTP findings are robust and provide an important avenue for future study. The experiments are performed carefully and the analysis is appropriate. The paper is well-written and the figures are clear.

We express our gratitude to Reviewer #1 for conducting a thoughtful and comprehensive review of our manuscript. We sincerely value both the constructive criticisms provided and your acknowledgment of the manuscript's strengths.

Weaknesses:The main weakness of the paper is the lack of causal manipulations to determine whether prevention or augmentation of any of the findings has any impact on behavior or cognition. Alternatively, if other manipulations would enhance working memory in ELS animals, it would be interesting to see the effects on any of these parameters measured in the paper.

We sincerely appreciate the insightful comments from Reviewer #1 regarding the potential benefits of including causal manipulations in our study. We wholeheartedly agree that such manipulations can provide a deeper understanding of the mechanistic underpinnings of the observed relationships and represent a crucial next step in our research trajectory.

Our primary objective in this study was to establish a comprehensive framework through observational examinations, exploring intricate relationships across various neurobiological and behavioral variables in the aftermath of early-life seizures (ELS). By identifying these associations, our work aims to provide a foundation for future investigations that can delve into targeted interventions.

While we acknowledge the importance of causal manipulations, we would like to underscore the advantages of our initial multivariate correlational study. Importantly, developmental brain disturbances have lasting impacts affecting multiple biological outcomes that may have intricate relationships between themselves. Firstly, although some neurobiological variables stood out from the comparisons of group means, this did not reveal some nuanced relationships within the data. The complexity of the relationships we uncovered, involving behavior, cognition, immunohistochemistry, plasticity, neurochemistry, and network dynamics, required a more elaborate analytical approach. Only through sophisticated data analysis techniques, we were able to dissect important peculiarities, such as the robust multivariate association between brain-wide astrogliosis and sensorimotor impairments, as well as non-linear relationships, such as the inverted-U relationship between plasticity and working memory. These nuances might not have been fully revealed through causal manipulations, since several variables are strongly related and consequently can affect several outcomes, leading to a false conclusion of direct causality.

Nevertheless, we acknowledge the understatement of the limitation of lacking causal manipulations in our manuscript. To address this, we have included a dedicated section in the discussion highlighting this limitation. We emphasize the advantages of this exploratory phase, supported by a review of the literature on cause-and-effect studies that align with our findings. Additionally, we speculate on promising targets for future cause-and-effect studies based on our findings. For instance, we hypothesize that enhancing plasticity may improve working memory in control subjects, while attenuating plasticity might have a similar effect in ELS subjects. Furthermore, we propose that reactive astrogliosis and concurrent neuroinflammatory processes likely underlie sensorimotor changes in the ELS group. Lastly, we suggest that dopaminergic antagonism in the ELS group could normalize behavioral deficits, prevent the exaggerated LTP induction of the HPC-PFC pathway, reestablish the state-dependent network dynamics, and desensitize the dopaminergic response.

[...]Also, I find the sections where correlations and dimensionality reduction techniques are used to compare all possible variables to each other less compelling than the rest of the paper (with the exception of the findings of U-shaped relationship of cognition to LTP). In fact, I think these sections take away from the impact of the actual findings.

We appreciate the reviewer's feedback and would like to emphasize the significance of the multivariate analysis conducted in our study. Multivariate analysis extends beyond bivariate correlations and is the only type of analysis capable of comprehending the relation of data in a multidimensional way, offering a comprehensive approach to understanding complex relationships among multiple variables. By employing techniques such as principal component analysis (PCA), generalized linear models (GLM), and canonical correlation analysis (CCA), we aimed to unravel intricate patterns of covariance that explore how different variables collectively contribute to the observed outcomes and assess the impact of each independent variable (predictor) on the dependent variable (the variable to be predicted or explained). Importantly, it enables us to control for potential confounding factors by keeping all other variables constant.

While we acknowledge that these sections may appear intricate, their inclusion is indispensable for a comprehensive understanding of the diverse variables associated with SE outcomes. We believe that these analyses offer valuable insights into the intricate dynamics of our study, providing a more holistic perspective on the altered spectrum induced by early-life seizures (ELS).

Regarding the reviewer's observations about the impact of the U-shaped relationship between cognition and LTP, we have made graphical and textual adjustments to emphasize the significance of these findings, aiming to enhance their clarity and impact within the broader context of our research. We trust that these modifications contribute to a more compelling presentation of our results.

[…]Finally, the apomorphine section seemed to hang separately from the rest of the paper and did not seem to fit well.

We appreciate the Reviewer #1 feedback on the apomorphine section. In order to address this point, we carefully rewrote our rationale before the results to clarify our hypothesis and chosen methodology. In our work, we performed the apomorphine experiment as a logical next step of previous data. We showed that ELS rats display REM-like oscillatory dynamics during active behavior, similar to genetically and pharmacologically hyperdopaminergic mice (Dzirasa et al., 2006). Furthermore, other results also indicated possible dopamine neurotransmission alterations, such as working memory deficits, hyperlocomotion, PPI deficits, aberrant HPC-PFC LTP, and abnormal PFC gamma coordination. Therefore, we hypothesized that ELS animals would present a state of hyperdopaminergic activity. Among the possible methodologies to investigate the hyperdopaminergic state, we choose the apomorphine sensitivity test, which is classically used and induces unambiguous behavior and neurochemical alterations in hyperdopaminergic rodents (Duval, 2023; Ellenbroek & Cools, 2002).

**Reviewer 1 (Recommendations For The Authors):**
(1) It would be useful to stain for other GABAergic interneuron markers such as somatostatin, VIP, CCK.(2) The authors refer to neuroinflammation but they are really referring to reactive astrogliosis. I would also suggest staining for microglial markers.(3) The duration of chronic electrographic seizures in ELS animals should also be calculated and presented.(4) Word usage: the authors frequently use the word "presents" when "demonstrates" would be more appropriate

(1) We appreciate your insight into staining for other GABAergic interneuron markers such as somatostatin, VIP, CCK. While investigating additional interneuron types is indeed relevant, it was not the primary focus of this study for several reasons: (1) The overall neuron density, assessed through NeuN immunostaining, revealed no differences between controls and early life seizure (ELS) groups, even in brain regions susceptible to neuron death after SE (i.e., CA1). Therefore, differences in interneurons, which are more resistant to death in SE and constitute approximately 20% of the cells, are unlikely. (2) Among all interneuron subtypes, Parvalbumin-positive (PV+) interneurons represent a substantial population and are susceptible to various stressors. In the hippocampus, 24% of GABAergic neurons are PV+, whereas 14% are SST+, 10% are CCK+, and VIP+ are less than 10% (Freund and Buzsaki, 1996). Consequently, we considered PV+ interneurons to be a more sensitive subpopulation for evaluating the effects of SE. As they showed no significant difference, we do not believe that assessing smaller subtypes, such as VIP+ or CCK+ cells, would yield significant differences.

(2) While we often see activated microglia in hippocampal sclerosis, these cells are only slightly increased in cases without hippocampal sclerosis (which are similar to our animals), as we previously published (Peixoto-Santos et al., 2012). Astrocytes are a better marker for the epileptogenic zone, as are increased in epileptogenic zones without neuron loss and are also important for controlling neuronal activity by neurotransmitter recycling and ion buffering. In fact, our present model is very similar to the mesial temporal lobe epilepsy patients with gliosis-only, which are characterized by only presenting increased reactive astrogliosis in the hippocampus, without cell loss, and also present changes in innate inflammatory response related to the presence of reactive astrocytes (Grote et al., 2023).

(3) We have performed these calculations and added this information to the revised manuscript.

(4) We thank the reviewer for the word usage recommendation. Indeed, we frequently used “present” throughout the manuscript to describe the observations and patterns the groups “exhibited” or “showed”. However, we believe this is truly not the most appropriate usage in the Discussion when we describe the multivariate latent factors, as we did not “present” them, but rather, we “demonstrated” their existence and significance through our analysis. We rewrote these sentences and hope this is the point the reviewer was referring to.

References:

Duval F. Systematic review of the apomorphine challenge test in the assessment of dopaminergic activity in schizophrenia. Healthcare. 2023 11 (1487): 1-11. doi: 10.3390/healthcare11101487.

Dzirasa K, Ribeiro S, Costa R, Santos LM, Lin SC, Grosmark A, Sotnikova TD, Gainetdinov RR, Caron MG, Nicolelis MAL. Dopaminergic control of sleep-wake states. Journal of Neuroscience. 2006 26:10577–10589. doi:10.1523/JNEUROSCI.1767-06.2006.

Freund TF, Buzsáki G. Interneurons of the hippocampus. Hippocampus. 1996;6(4):347-470. doi: 10.1002/(SICI)1098-1063(1996)6:4<347::AID-HIPO1>3.0.CO;2-I. PMID: 8915675.

Ellenbroek BA & Cools AR. Apomorphine susceptibility and animal models for psychopathology: genes and environment. Behavior Genetics. 2002 32 (5): 349-361. doi: 10.1023/a:1020214322065.

Grote A, Heiland DH, Taube J, Helmstaedter C, Ravi VM, Will P, Hattingen E, Schüre JR, Witt JA, Reimers A, Elger C, Schramm J, Becker AJ, Delev D. 'Hippocampal innate inflammatory gliosis only' in pharmacoresistant temporal lobe epilepsy. Brain. 2023 Feb 13;146(2):549-560. doi: 10.1093/brain/awac293. PMID: 35978480; PMCID: PMC9924906.

Peixoto-Santos JE, Galvis-Alonso OY, Velasco TR, Kandratavicius L, Assirati JA, Carlotti CG, Scandiuzzi RC, Serafini LN, Leite JP. Increased metallothionein I/II expression in patients with temporal lobe epilepsy. PLoS One. 2012;7(9):e44709. doi: 10.1371/journal.pone.0044709. Epub 2012 Sep 18. Erratum in: PLoS One. 2016;11(7):e0159122. PMID: 23028585; PMCID: PMC3445538.

**Reviewer 2**
In this manuscript, the authors employ a multilevel approach to investigate the relationship between the hippocampal-prefrontal (HPC-PFC) network and long-term phenotypes resulting from early-life seizures (ELS). Their research begins by establishing an ELS rat model and conducting behavioral and neuropathological studies in adulthood. Subsequently, the manuscript delves into testing hypotheses concerning HPC-PFC network dysfunction. While the results are intriguing, my enthusiasm is tempered by concerns related to the logical flow

We thank the reviewer for bringing attention to the logical flow of the manuscript. Given the diverse array of behavioral and neurobiological variables examined in our study obtained through various methods and measures, we utterly recognize the utmost importance of a clear and coherent logical flow to provide a comprehensive understanding of the overall narrative.

Our goal was to articulate the neurobiological findings in a manner that underscores their convergence of mechanisms, revealing a cohesive relationship between early-life seizure, cognitive deficits, sensorimotor impairments, abnormal network dynamics, aberrant plasticity, neuroinflammation and dysfunctional dopaminergic transmission.

Briefly, an outline of our narrative could be summarized in the highlights:

(1) ELS induces sensorimotor alterations and working memory deficits.

(2) ELS does not induce neuronal loss, so neurobiological underpinnings may be molecular and functional.

(3) ELS induces brain-wide astrogliosis and exaggerated HPC-PFC long-term plasticity.

(4) ELS does not induce neuronal loss, so neurobiological underpinnings may be molecular and functional.

(5) Sensorimotor alterations are more correlated to astrogliosis, while cognitive deficits to altered HPC-PFC plasticity.

(6) ELS-induced functional alterations may also be observable in freely moving subjects. ELS induces state-dependent alterations in the HPC-PFC network dynamics, such as increased hippocampal theta and abnormal PFC gamma coordination during behavioral activity.

(7) ELS leads to REM-ACT similarity, previously reported in hyperdopaminergic mice, indicating dopaminergic dysfunction.

(8) ELS exhibits altered dopaminergic transmission and behavioral sensitivity that mirror the initial sensorimotor findings.

(9) The literature establishes an inverted-U relationship between dopamine and cognition and PFC plasticity, which may explain our finding of an inverted-U relationship between working memory and HPC-PFC LTP across CTRL and ELS rats.

To address this concern, we have made revisions to enhance the logical flow, ensuring a more seamless transition between the different sections of the Results by presenting clearer links between observations and following investigations. We hope these changes contribute to a more straightforward rationale and easily understandable presentation of our hypotheses and results.

Focus on Correlations: The manuscript primarily highlights correlations as the most significant findings. For instance, it demonstrates that ELS induces cognitive and sensorimotor impairments. However, it falls short of elucidating why these deficits are specifically linked to HPC-PFC synaptic plasticity/network. Furthermore, the manuscript mentions the involvement of other brain regions like the thalamus in the long-term outcomes of ELS based on immunohistochemistry data.

Thank you for your insightful comments, which allowed us to provide further clarification on our study's focus and findings. Our primary goal was to delve into the electrophysiological alterations within the HPC-PFC pathway. The rationale behind this choice lies in the hypothesis that, even in the absence of significant neuronal loss, functional changes in circuits closely linked to the cognitive and behavioral aspects under investigation could be identified.

While we concentrated our electrophysiological investigation on the HPC-PFC pathway due to its well-established functional correlates in existing literature, it is essential to highlight that our data reveal broader alterations in neural circuitry. Notably, we observed an increase in GFAP in the entorhinal cortex and thalamic reticular nucleus, along with changes in the dopaminergic release within the VTA-NAc pathway. These findings suggest that the impact of early-life seizures extends beyond the HPC-PFC circuit.

While we recognize the relevance of other brain circuits in the outcomes of ELS, we argue for a specific role of the HPC-PFC circuit in the outcomes of ELS. We will detail the supporting evidence and arguments that specifically link the HPC-PFC function to our ELS-related observations in a later comment regarding the "overinterpretation" of the HPC-PFC role.To better convey these important nuances, we have made specific modifications to the results and in the discussion section to underscore the broader implications of our findings, providing a more comprehensive understanding of the study's scope and outcomes.

[…]This raises questions about the subjective nature and persuasiveness of the statistical studies presented.

All statistical analyses were carefully applied based on the literature and following well-established precepts and precautions. Specifically, we constructed the experimental design for univariate inferential statistics for the data related to behavioral tests, synaptic plasticity, immunohistochemistry, oscillatory activity, and dopaminergic sensitization. However, we also submitted our data to multivariate statistical analysis, which is recommended in cases with a considerable amount of data, and intend to investigate possible hidden effects. In this situation, multivariate analyses are inherently exploratory due to the possibility of using multiple measurements for each phenomenon investigated. Nevertheless, their application is not subjective and follows the same statistical rigor as univariate analyses. We firmly believe that abstaining from exploring these data, would not reach the full potential of this analytical method in dissecting the multidimensional associations within our dataset. In order to eliminate any doubt regarding the objectivity in the choice and application of statistics, we carefully rewrote the methods, highlighting the details of statistical rigor even more.

Sample Size Concerns: The manuscript raises concerns about the adequacy of sample sizes in the study. The initial cohort for acute electrophysiology during ELS induction comprised only 5 rats, without a control group. Moreover, the behavioral tests involved 11 control and 14 ELS rats, but these same cohorts were used for over four different experiments. Subsequent electrophysiology and immunohistochemistry experiments used varying numbers of rats (7 to 11). Clarification is needed regarding whether these experiments utilized the same cohort and why the sample sizes differed. A power analysis should have been performed to justify sample sizes, especially given the complexity of the statistical analyses conducted.

We appreciate the reviewer's thoroughness and considerations regarding the sample sizes used in our study. The concerns raised about statistical robustness seem to stem from a lack of clarity in delineating the rat cohorts used in each experiment. It is encouraging to note that several studies in the field of neurophysiology, employing similar analyses, utilize a sample size similar to what was used in our research. The choice of the sample size was based on a thorough analysis of the existing literature, considering specific experimental demands, the complexity of employed techniques, and the need to achieve statistically robust results. In response to these concerns and to enhance clarity on the sample sizes, we have made several modifications (highlighted in red) in the text. Below, we provide details for each animal cohort utilized:

Cohort 1 - Acute Electrophysiology

The decision to use only 5 animals without a control group for acute electrophysiological recording aimed specifically to confirm that the injection of lithium-pilocarpine would induce both behavioral and electrographic seizures. It is crucial to note that this was a descriptive result and a methodological control of the ELS model. Besides, no statistical test or further analysis was conducted on these data. We maintain the belief that a group of 5 animals is sufficient to demonstrate that the protocol induces electrographic seizures, and introducing a control group was considered unnecessary to show that saline injection does not induce electrographic seizures.

Cohort 2 - Behavior, LTP Recording, and Immunohistochemistry

Initially, 14 (ELS) and 11 (CTRL) rats were used for behavior assessment. The reduction in sample size for LTP and immunohistochemistry experiments was influenced by practical challenges, including mortality during LTP surgery and issues with immunohistochemical staining that hindered a proper analysis for some animals.

Cohort 3 - Chronic Freely-Moving Electrophysiology

A new cohort of animals (n=6 and 9 for CTRL and ELS, respectively) was used specifically for freely-moving electrophysiological data.

Cohort 4 - Behavioral Sensitization to Psychostimulants

A fourth cohort was utilized for assessing behavioral sensitization to psychostimulants (CTRL n=15 and ELS n=14). The reduced sample size for neurotransmitter analysis (CTRL n=8 and ELS n=9) was a deliberate selection of a subsample to ensure a sufficient sample for quantification while maintaining statistical validity

Overinterpretation of HPC-PFC Network Dysfunction: The manuscript potentially overinterprets the role of HPC-PFC network dysfunction based on the results.

We appreciate the insight from Reviewer #2 regarding the potential overinterpretation of the role of the hippocampal-prefrontal cortex (HPC-PFC) network dysfunction in the various alterations observed after ELS.

The significance of HPC-PFC plasticity and network function has been extensively documented concerning cognitive, affective, and sensorimotor functions, as well as in models of neuropsychiatric diseases. Our recent review (Ruggiero et al., 2021) compiles these findings. Specifically, the HPC-PFC network has been linked to spatial working memory through a series of causal and correlational studies conducted by Floresco et al. and Gordon et al. These findings make the HPC-PFC pathway a plausible candidate for underlying alterations associated with working memory, consistent with our observation of exaggerated HPC-PFC LTP associated with poorer performance in the ELS group. Regarding the immunohistochemical observations, we concur with Reviewer #2 that these findings suggest broader-scale brain alterations related to sensorimotor dysfunction beyond the HPC-PFC circuitry. Surely, we acknowledge that these large-scale alterations may underlie brain-wide network functional changes.

In our network dynamics study arm, we investigated HPC-PFC oscillatory activity, allowing us to discuss potential relationships between abnormal plasticity (verified in the second study arm) and network dynamics. It is important to note that while there is some anatomical specificity to the LFPs recorded in the HPC and PFC, these activities may represent larger-scale limbic-cortical dynamics. The intermediate HPC exhibits a significant influence from both dorsal and ventral HPC, and the prelimbic PFC is intricately related to both hippocampal and thalamic oscillations exhibiting under-demand state-dependent synchrony. Additionally, the state maps used in our study were initially described to distinguish states at a global forebrain network level. Even in our past studies, we have described HPC-PFC patterns of network activity (Marques et al., 2022a) that later were found to represent a part of a brain-wide synchrony pattern (Marques et al., 2022b). However, most of our findings on oscillatory dynamics were centered around theta oscillations, a well-established brain-wide activity that originates and spreads from the hippocampus and are present in the HPC-PFC circuit during activity.

In conclusion, we believe the correlations between HPC-PFC LTP and working memory, as well as the specific alterations of theta coordinated activity, support a particular role of the HPC-PFC network dysfunction in the effects of ELS. However, the brain-wide immunochemical alterations are plausible indications of larger-scale dysfunctional networks. To address this issue, we emphasized in the discussion of network findings that the immunohistochemical and neurochemical findings endorse the need to investigate ELS effects on larger networks.

Notably, cognitive deficits are described as subtle, with no evidence of learning deficits and only faint working memory impairments. However, sensorimotor deficits show promise. Consequently, it's essential to justify the emphasis on the HPC-PFC network as the primary mechanism underlying ELS-associated outcomes, especially when enhanced LTP is observed. Additionally, the manuscript seems to sideline neuropathological changes in the thalamus and the thalamus-to-PFC connection. The analysis lacks a direct assessment of the causal relationship between HPC-PFC dysfunction and ELS-associated outcomes, leaving a multitude of multilevel analyses yielding potential correlations without easily interpretable results.

We thank Reviewer #2 for the thorough review and insightful comments. To better grasp the context, it is crucial to consider this characterization within the scope of our experimental design and expected outcomes. Unlike epilepsy models involving adult animals or interventions causing pronounced neuronal loss and structural modifications, our study was intentionally designed to explore moderate behavioral alterations. In fact, the mild behavioral alterations observed in ELS models and the lack of neuronal loss guided our focus on investigating changes in HPC-PFC communication.

While our observed cognitive deficits may be milder compared to certain models, it is imperative to underscore their robustness and clinical relevance. These findings have been consistently replicated globally across various experimental models, encompassing ELS induced by hyperthermia (Chang et al., 2003; Kloc et al., 2022), kainic acid (Statsfrom et al. 1993), flurothyl (Karnam et al., 2009a; 2009b), and hypoxia (Najafian et al., 2021; Hajipour et al., 2023). Mild cognitive deficits were also evident by other research groups using the pilocarpine model in P12 (Mikulecká et al., 2019; Kubová et al., 2013; Kubová et al., 2002). Furthermore, our group replicated the working memory deficit results using an alternative paradigm (the T-maze) and a different rat strain (Sprague Dawley), enhancing the reliability of our observations (D’Agosta et al., 2023).

The clinical perspective gains importance, considering that cognitive effects of ELS may be less severe than those in patients with long-term epilepsy. In fact, the majority of patients with childhood epilepsy exhibit mild cognitive impairment as the most common grade of severity - more than two times the rate of severe cognitive impairment (Sorg et al., 2022). Investigating the mechanisms underlying these mild cognitive changes is crucial for shedding light on neurobiological aspects not fully understood, thereby expanding our comprehension of the consequences of ELS.

We recognize the challenges associated with conducting causal experiments in neuroscience, especially in long-term and chronic alterations as seen in our model. Isolating modifications of specific activities is indeed intricate. However, it's essential to acknowledge that neuroscience progress has not solely relied on causal experiments but has significantly advanced through correlational observations. Our findings serve as a foundational step in comprehending the repercussions of ELS, proposing mechanisms and circuits that necessitate further in-depth dissection and study in the future. We have integrated these considerations into the discussion section of the manuscript to enhance clarity.

Overall, while the manuscript presents intriguing findings related to the HPC-PFC network and ELS outcomes, it requires a more rigorous experimental design[…]

We thank the reviewer for acknowledging our intriguing findings. Regarding the experimental design, we are confident that all the manuscript hypotheses, design, and execution of experiments were rigorously based on the literature and carried out with all necessary controls. As stated earlier, we constructed the experimental design for univariate inferential statistics and explored associations between variables using multivariate statistics. Specifically, we achieved a rigorously experimental design following a series of guidelines. First, the planning of the sample size in each experiment and their respective controls were based on mild effects from the ELS literature. As previously indicated, the only experiment with one group was just the description of the behavioral effects and electrographic seizures after the acute injection of lithium-pilocarpine. Given the exhaustive replication of these data in the ELS literature, this result was presented descriptively as a methodological control. Second, detailed descriptions of statistics were made in both methods and results, always indicating positive and negative results. Notably, the experimental designs used in the work do not correspond to any novelty or radicalization, strictly following the literature of the field. However, new indications and references about the experimental accuracy were added to the manuscript to resolve any doubts regarding objectivity.

References:

Chang YC, Huang AM, Kuo YM, Wang ST, Chang YY, Huang CC. Febrile seizures impair memory and cAMP response-element binding protein activation. Ann Neurol. 2003 Dec;54(6):706-18. doi: 10.1002/ana.10789. PMID: 14681880.

D'Agosta R, Prizon T, Zacharias LR, Marques DB, Leite JP, Ruggiero RN. Alterations in hippocampal-prefrontal cortex connectivity are associated with working memory impairments in rats subjected to early-life status epilepticus. In: NEWROSCIENCE INTERNATIONAL SYMPOSIUM, 2023, Ribeirão Preto. Poster.

Hajipour S, Khombi Shooshtari M, Farbood Y, Ali Mard S, Sarkaki A, Moradi Chameh H, Sistani Karampour N, Ghafouri S. Fingolimod Administration Following Hypoxia Induced Neonatal Seizure Can Restore Impaired Long-term Potentiation and Memory Performance in Adult Rats. Neuroscience. 2023 May 21;519:107-119. doi: 10.1016/j.neuroscience.2023.03.023. Epub 2023 Mar 28. PMID: 36990271.

Karnam HB, Zhou JL, Huang LT, Zhao Q, Shatskikh T, Holmes GL. Early life seizures cause long-standing impairment of the hippocampal map. Exp Neurol. 2009 Jun;217(2):378-87. doi: 10.1016/j.expneurol.2009.03.028. Epub 2009 Apr 2. PMID: 19345685; PMCID: PMC2791529.

Karnam HB, Zhao Q, Shatskikh T, Holmes GL. Effect of age on cognitive sequelae following early life seizures in rats. Epilepsy Res. 2009 Aug;85(2-3):221-30. doi: 10.1016/j.eplepsyres.2009.03.008. Epub 2009 Apr 22. PMID: 19395239; PMCID: PMC2795326.

Kubová H, Mareš P. Are morphologic and functional consequences of status epilepticus in infant rats progressive? Neuroscience. 2013 Apr 3;235:232-49. doi: 10.1016/j.neuroscience.2012.12.055. Epub 2013 Jan 7. PMID: 23305765.

Kloc ML, Marchand DH, Holmes GL, Pressman RD, Barry JM. Cognitive impairment following experimental febrile seizures is determined by sex and seizure duration. Epilepsy Behav. 2022 Jan;126:108430. doi: 10.1016/j.yebeh.2021.108430. Epub 2021 Dec 10. PMID: 34902661; PMCID: PMC8748413.

Kubová H, Mares P, Suchomelová L, Brozek G, Druga R, Pitkänen A. Status epilepticus in immature rats leads to behavioural and cognitive impairment and epileptogenesis. Eur J Neurosci. 2004 Jun;19(12):3255-65. doi: 10.1111/j.0953-816X.2004.03410.x. PMID: 15217382.

Marques DB, Ruggiero RN, Bueno-Junior LS, Rossignoli MT, and Leite JP. Prediction of Learned Resistance or Helplessness by Hippocampal-Prefrontal Cortical Network Activity during Stress. The Journal of Neuroscience. 2022a 42 (1): 81-96.. https://doi.org/10.1523/jneurosci.0128-21.2021.

Marques DB, Rossignoli MT, Mesquita BDA, Prizon T, Zacharias LR, Ruggiero RN and Leite JP. Decoding fear or safety and approach or avoidance by brain-wide network dynamics abbreviated. bioRxiv. 2022b https://doi.org/10.1101/2022.10.13.511989.

Mikulecká A, Druga R, Stuchlík A, Mareš P, Kubová H. Comorbidities of early-onset temporal epilepsy: Cognitive, social, emotional, and morphologic dimensions. Exp Neurol. 2019 Oct;320:113005. doi: 10.1016/j.expneurol.2019.113005. Epub 2019 Jul 3. PMID: 31278943.

Najafian SA, Farbood Y, Sarkaki A, Ghafouri S. FTY720 administration following hypoxia-induced neonatal seizure reverse cognitive impairments and severity of seizures in male and female adult rats: The role of inflammation. Neurosci Lett. 2021 Mar 23;748:135675. doi: 10.1016/j.neulet.2021.135675. Epub 2021 Jan 28. PMID: 33516800.

Ruggiero RN, Rossignoli MT, Marques DB, de Sousa BM, Romcy-Pereira RN, Lopes-Aguiar C and Leite JP. Neuromodulation of Hippocampal-Prefrontal Cortical Synaptic Plasticity and Functional Connectivity: Implications for Neuropsychiatric Disorders. Frontiers in Cellular Neuroscience. 2021 15 (October): 1–23. https://doi.org/10.3389/fncel.2021.732360.

Sorg AL, von Kries R, Borggraefe I. Cognitive disorders in childhood epilepsy: a comparative longitudinal study using administrative healthcare data. J Neurol. 2022 Jul;269(7):3789-3799. doi: 10.1007/s00415-022-11008-y. Epub 2022 Feb 15. PMID: 35166927; PMCID: PMC9217877.

Stafstrom CE, Chronopoulos A, Thurber S, Thompson JL, Holmes GL. Age-dependent cognitive and behavioral deficits after kainic acid seizures. Epilepsia. 1993 May-Jun;34(3):420-32. doi: 10.1111/j.1528-1157.1993.tb02582.x. PMID: 8504777.